# In Context Semi-Supervised Learning

**Jiashuo Fan**[*]
Department of Electrical and Computer Engineering
Duke University
jf381@duke.edu

**Paul Rosu**[*]
Department of Electrical and Computer Engineering
Duke University
pr145@duke.edu

**Aaron T. Wang**
Department of Electrical and Computer Engineering
Duke University
tw231@duke.edu

**Lawrence Carin**
Department of Electrical and Computer Engineering
Duke University
lcarin@duke.edu

**Xiang Cheng**
Department of Electrical and Computer Engineering
Duke University
xiang.cheng@duke.edu

## Abstract

There has been significant recent interest on understanding the capacity of Transformers for in-context learning (ICL), yet most theory focuses on supervised settings with explicitly *labeled* pairs. In practice, Transformers often perform well even when labels are sparse or absent, suggesting crucial structure within unlabeled contextual demonstrations. We introduce and study *in-context semi-supervised learning (IC-SSL)*, where a small set of labeled examples is accompanied by many unlabeled points, and show that Transformers can leverage the unlabeled context to learn a robust, context-dependent representation. This representation enables accurate predictions and markedly improves performance in low-label regimes, offering foundational insights into how Transformers exploit unlabeled context for representation learning within the ICL framework. Our code is available at https://github.com/Jason-fan20/ICL_Semi.

## 1 Introduction

The Transformer (Vaswani et al., 2017) has revolutionized natural language processing, achieving remarkable success in tasks like language generation Vaswani et al. (2017); Radford et al. (2019); Devlin et al. (2019); Brown et al. (2020); Touvron et al. (2023); DeepSeek (2025); it has also been applied in many other areas, including vision Dosovitskiy et al. (2021), network analysis Cheng et al. (2025); Kreuzer et al. (2021); Rampasek et al. (2022) and biology Jiang et al. (2024). Motivated by the demonstrated effectiveness of Transformers for few-shot (or in-context) learning Brown et al. (2020), a promising line of recent research has focused on functional data von Oswald et al. (2023); Ahn et al. (2023); Zhang et al. (2023); Schlag et al. (2021). In this setting one is given a set of contextual data pairs, where each pair is characterized by the observed input and output of a *latent* function. It has been demonstrated that a properly designed Transformer can in its forward pass perform functional gradient descent (GD) – or close variations thereof – for the function, and it can use that inference to predict the expected functional outcome for a specified query input to the function. The Transformer can perform this inference in-context, with no parameter fine-tuning.

Early work on Transformer-based in-context learning (ICL) focused on linear functions von Oswald et al. (2023); Ahn et al. (2023); Zhang et al. (2023); Schlag et al. (2021), and that has been extended recently to nonlinear functions that reside in a reproducing kernel Hilbert space (RKHS) Cheng et al. (2024). These works focus on learning real-valued outcomes, and similar analysis was recently developed for categorical outcomes Wang et al. (2025). Categorical observations are of interest when studying ICL for classification problems.

---

[*]Equal contribution

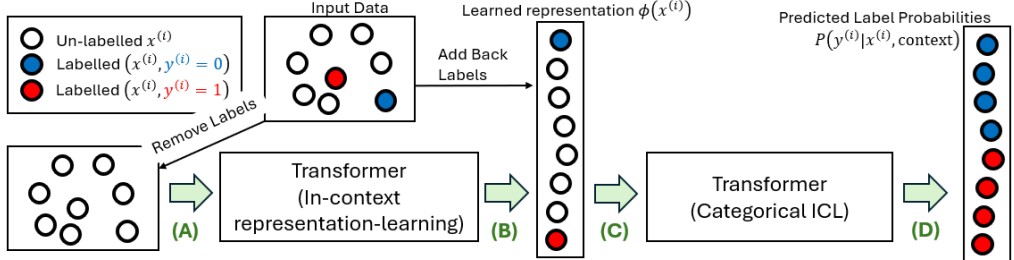

Figure 1: Setup for semi-supervised in-context learning with a Transformer. Input Data consists of labeled pairs $\{(x^{(1)}, y^{(1)}), \ldots, (x^{(m)}, y^{(m)})\}$, where $y^{(i)} \in \{0, 1\}$, and unlabeled data $\{x^{(m+1)}, \ldots, x^{(n)}\}$. In stage (A), a Transformer module takes as input $\{x^{(1)}, \ldots, x^{(n)}\}$ and outputs $\phi(x^{(i)})$, where $\phi(\cdot)$ is a **context-dependent feature representation**. In stage (B), we augment $\{\phi(x^{(1)}), \ldots, \phi(x^{(n)})\}$ with the $m$ *observed labels*. In stage (C), a second Transformer module takes this augmented set as input. In stage (D), the second Transformer module outputs prediction probabilities of $y^{(m+1)}, \ldots, y^{(n)}$ for each unlabeled $x^{(i)}$ in standard ICL fashion. The two Transformer modules combine into a single Transformer, trained end-to-end. As detailed in Section 3, the first module is motivated by learning an eigenmap of the Laplacian, and the second module performs ICL with categorical observations by implementing gradient descent at inference time.

Prior work on Transformer-based ICL largely treats the supervised case, using contextual *labeled* pairs. When unlabeled points appear, their labels are inferred independently, ignoring structure in the full unlabeled set. Yet in practice we have many unlabeled samples and few labels, and so it is of interest for ICL to leverage the context provided by *all* of this data.

## 1.1 OUTLINE OF CONTRIBUTIONS

**In-context semi-supervised learning setup.** We extend Transformer-based in-context learning to a semi-supervised setting, broadening the concept of "context" to leverage both labeled and unlabeled data within a unified end-to-end Transformer architecture (Figure 1). We demonstrate how Transformers can effectively perform semi-supervised ICL, using scarce labeled data alongside abundant unlabeled examples, without the need for separate preprocessing of inputs.

**A two-stage Transformer construction for representation learning + ICL.** We construct a two-stage Transformer (Section 3). The first stage performs representation learning from unlabeled data by forming a discrete Laplacian from local Euclidean distances, then computing an Eigenmap; this yields context-dependent manifold structure in the forward pass (Section 3.1). The second stage (Section 3.2) is a construction for a Transformer-based ICL module that can be shown to perform in-context learning for categorical observations by implementing functional gradient descent in its forward pass. *Mechanistically*, each stage is capable of implementing an explicit algorithm (1. Laplacian construction + power iteration; 2. kernelized GD for in-context inference), providing a first step toward a transparent account of how attention and MLPs realize semi-supervised ICL computations. In practice, we instantiate both stages inside a single Transformer and train the model end-to-end with the IC-SSL objective in Eq. (4); the modular construction serves as a strong mechanistic inductive bias/initialization rather than a hard constraint, and joint optimization can deviate from and improve upon the strict two-stage algorithm when beneficial. In addition, the construction serves as a *prototype* for analyzing full end-to-end Transformers: it gives a modular reference representation against which we can compare baseline Transformers trained directly on data (e.g., via separation and alignment analyses in Section F.6).

**Mechanistic understanding of IC-SSL.** Section F.6 examines our method across manifolds of increasing complexity — low-dimensional manifolds in $\mathbb{R}^3$ (Figure 2), higher-dimensional product manifolds in $\mathbb{R}^{15}$, and a high-dimensional image manifold from Stable Diffusion v1.5 (Figure 5) — where our construction consistently outperforms strong baselines. We then use the construction as a *reference model* to interpret how an unconstrained, end-to-end Transformer learns to perform in-context semi-supervised learning on real data (ImageNet100). Our model serves as a reference point for what representations are learned by the standard Transformer (Table 1), as well as for understanding the phase-transitions in the standard Transformer's learning curves (Figure 3).

**Inductive bias and learning.** Our two-stage design imposes a concrete inductive bias—local, sparse attention that can effectively implement spectral feature learning (first stage) and gradient-based learning (second stage) This makes IC-SSL markedly more sample-efficient in low-data regimes. As shown in Section F.6, across elementary manifolds in $\mathbb{R}^3$ (Figure 2), higher-dimensional product manifolds in $\mathbb{R}^{15}$ (Figure 4c), and high-dimensional image manifolds (Figure 5), this bias

translates into consistently higher accuracy than strong baselines, including variants that rely on offline Laplacian Eigenmaps, even with few labeled examples. The same structural bias also supports robust generalization: the model retains strong in-distribution performance and transfers to previously unseen geometries (Figure 4b) and modalities (Figure 5), indicating that it learns geometry-aware computations rather than overfitting to a specific dataset or coordinate system.

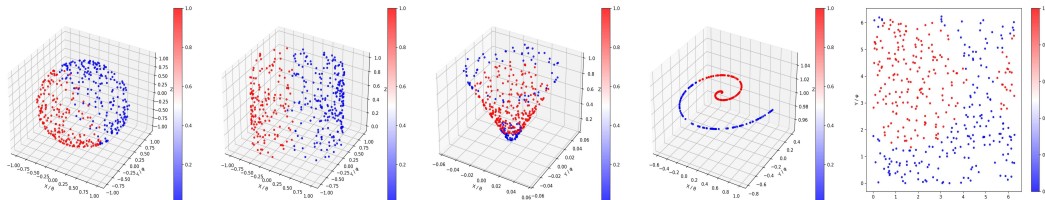

Figure 2: Synthetic manifolds we use. Left to right: Sphere ($\mathbb{S}^2$), Right Circular Cylinder (C), Right Circular Cone ($\text{Cone}_\alpha$), Archimedean Spiral (Swiss-Roll, SR), Flat Torus ($\mathbb{T}^2$). The colors reflect the binary labels.

## 1.2 RELATED WORK

Semi-supervised learning (SSL) has been a problem of long-standing interest to the machine learning community van Engelen & Hoos (2020); Ouali et al. (2020); Chapelle et al. (2010). It is particularly valuable when labeled data are scarce, multiple unlabeled samples are observed at once, and there is an underlying structure associated with the covariates from both labeled and unlabeled data. Despite the recent surge in research on Transformer-based In-Context Learning (ICL) von Oswald et al. (2023); Ahn et al. (2023); Zhang et al. (2023); Schlag et al. (2021); Cheng et al. (2024); Wang et al. (2025), standard implementations often overlook the potential of the unlabeled portion of the context.

Recent work has begun to investigate this intersection, primarily focusing on scaling and data selection. Agarwal et al. (2024) demonstrated that scaling to many-shot prompts yields significant gains and introduced Unsupervised ICL (using inputs only) to reduce dependence on curated labels. Similarly, Chen et al. (2025) improved many-shot ICL by influence-selecting unlabeled examples and pseudo-labeling them with an LLM. From a theoretical perspective, Li et al. (2025) proved that while a single-layer linear-attention model cannot exploit unlabeled data, depth or looping constructs can implement polynomial estimators akin to the Expectation-Maximization (EM) algorithm, showing empirical gains on tabular tasks.

These findings motivate our focus on the fundamental mechanism by which Transformers leverage unlabeled context. Unlike previous approaches that rely on adding pseudo-labeled shots Chen et al. (2025); Agarwal et al. (2024), we argue that Transformers can extract structure directly from unlabeled tokens. This relates to the field of spectral geometry and graph-based embeddings. Traditionally, embedding vectors for unlabeled samples—often derived from the eigenvectors of the graph Laplacian Coifman & Lafon (2006)—have been calculated offline and used as positional embeddings (PEs) in graph Transformers Kreuzer et al. (2021); Rampasek et al. (2022). While Cheng et al. (2025) recently showed that Transformers can compute certain PEs during inference, we are unaware of any prior work that fully integrates this into a semi-supervised ICL framework.

In this work, we bridge this gap by demonstrating that a full Transformer extracts geometry-aligned representations in-context. By computing a representation from unlabeled tokens and using subsequent attention layers to implement a gradient-based learning algorithm, our approach provides an explicit forward-pass mechanism that explains why depth is critical in semi-supervised ICL. This mechanism is supported by cross-domain evidence—ranging from synthetic manifolds to diffusion-generated images and ImageNet100 features—showing that the Transformer learns representations consistent with the underlying geometry of the data.

## 2 PROBLEM SETUP

**Background: In-Context Supervised Learning.** For in-context *supervised* learning von Oswald et al. (2023); Cheng et al. (2024), the Transformer is presented with contextual data $\mathcal{C} = \left\{(x^{(1)}, y^{(1)}), \ldots, (x^{(m)}, y^{(m)})\right\}$, where $x^{(i)} \in \mathbb{R}^d$, and $y^{(i)}$ is a label. The query is $x^{(m+1)}$. The goal of prior work von Oswald et al. (2023); Cheng et al. (2024) was examination of the Transformer to infer $\mathbb{E}(y^{(m+1)})$ based on the contextual data. In this setting, it was assumed that there is an underlying function $f(x)$ responsible for the relationship between any input $x^{(i)}$ and output $y^{(i)}$. In

most prior work of this type, investigation was performed on the potential for Transformers to infer context-dependent $f(x)$ on the forward pass, and use it to estimate $\mathbb{E}(y^{(m+1)})$.

In most prior work von Oswald et al. (2023); Cheng et al. (2024) the outcomes $y^{(i)}$ were real. Recently this has been extended to categorical $y^{(i)} \in \{1, \ldots, C\}$ Wang et al. (2025), and we will consider that setting here. Given $f(x)$, the relationship between $x^{(i)}$ and $y^{(i)}$ is modeled via the softmax (like at the output of language models Vaswani et al. (2017)):

$$\mathbb{P}\Big(y^{(i)}|x^{(i)}\Big) = \frac{\exp\left(w_{y^{(i)}}^\top f(x^{(i)})\right)}{\sum_{c=1}^{C} \exp\left(w_c^\top f(x^{(i)})\right)}. \tag{1}$$

For each category (label type) $c$ there is a learned embedding vector $w_c \in \mathbb{R}^{d'}$, and the function $f(x)$ has a $d'$-dimensional vector output. In-context learning (ICL) in this setting was first developed in Wang et al. (2025), and for reader convenience details are provided in Appendix B.

A limitation of this approach is that the context that is exploited is only manifested in the form of the observed labeled data, and each query is analyzed in isolation. A contribution of this paper concerns extending this concept to a *semi-supervised ICL* setting, as discussed next.

**In-context *semi-supervised* learning.** Let $n$ denote the total number of points, and let $m < n$ denote the number of labeled points. The input to the Transformer is thus

$$\mathcal{C} = \underbrace{\left\{(x^{(1)}, y^{(1)}), \ldots, (x^{(m)}, y^{(m)})\right\}}_{\text{labeled points}} \cup \underbrace{\left\{x^{(m+1)}, \ldots, x^{(n)}\right\}}_{\text{unlabeled points}}. \tag{2}$$

The goal is to predict the expected values of $y^{(m+1)}, \ldots, y^{(n)}$, with this prediction placed within the context of *all* data, all $n$ observed covariates and the $m$ labels.

Define the matrix $X = (x^{(1)}, \ldots, x^{(n)}) \in \mathbb{R}^{d \times n}$, and the function $\phi(X)$ that outputs a matrix $\Phi = (\phi^{(1)}, \ldots, \phi^{(n)})$, with $\phi^{(i)}$ a feature vector for $x^{(i)}$. Importantly, $\phi(X)$ is a function of *all* observed covariates, and therefore features $\phi^{(i)}$ are not only dependent on $x^{(i)}$, but also account for context (e.g., manifold information) provided by the columns of $X$. The model in (1) generalizes to

$$\mathbb{P}\Big(y^{(i)}|X\Big) = \exp\left(w_{y^{(i)}}^\top f(\phi^{(i)})\right) / \sum_{c=1}^{C} \exp\left(w_c^\top f(\phi^{(i)})\right). \tag{3}$$

Note that the observation $y^{(i)}$ associated with any sample is now dependent on *all* covariates, via the function $\phi(X)$. Further, the ICL prediction we develop in Section 3.2 is based on all observed labels $y^{(1)}, \ldots, y^{(m)}$; thus, cumulatively, the setup will utilize all contextual data, labeled and unlabeled.

## 2.1 IN-CONTEXT SEMI-SUPERVISED LOSS

We will define two Transformers, $\mathsf{TF}_{rep}$ and $\mathsf{TF}_{sup}$, which are respectively responsible for inferring $\phi$ (representation learning), and subsequently inferring $f$. Architectural details of $\mathsf{TF}_{rep}$ and $\mathsf{TF}_{sup}$ are discussed in Section 3 below. Let $\{\theta_{rep}, \theta_{sup}\}$ denote parameters for $\{\mathsf{TF}_{rep}, \mathsf{TF}_{sup}\}$. Let $\theta = \theta_{rep} \cup \theta_{sup} \cup \{w_c\}_{c=1,\ldots,C}$ denote the union of all parameters. Then given a context (2) of encompassing labeled and unlabeled samples, the *in-context semi-supervised cross-entropy loss* is:

$$L(\theta; \mathcal{C}) = \frac{1}{n-m} \sum_{i=m+1}^{n} \log\left(\frac{\exp\left(w_{y^{(i)}}^\top f(\phi(x^{(i)}))\right)}{\sum_{c=1}^{C} \exp\left(w_c^\top f(\phi(x^{(i)}))\right)}\right). \tag{4}$$

When training, this loss is fit to the $n - m$ *unlabeled* samples (for which we have labels *when training*). The training objective of the Transformer is thus $\min_\theta \mathbf{L}(\theta)$, where $\mathbf{L}(\theta) = \mathbb{E}_{\mathcal{C}}[L(\theta; \mathcal{C})]$, i.e., an average over the available training contextual datasets. Note that the optimization is performed wrt the parameters $\theta$, not *directly* on the features $\phi^{(i)}$ and function $f(\phi^{(i)})$, which are manifested on the Transformer forward pass.

## 3 TRANSFORMER CONSTRUCTION

### 3.1 TRANSFORMER FOR IN-CONTEXT REPRESENTATION LEARNING

Given $\mathcal{C}$, We define the affinity matrix $\mathcal{A} \in \mathbb{R}^{n \times n}$ with RBF bandwidth $h$ as $\mathcal{A}_{i,j} = \exp(-\|x^{(i)} - x^{(j)}\|_2^2/(2h))$, and $\mathcal{A}_{i,i} := 0$. Let $\mathcal{D}$ be the diagonal matrix defined as $\mathcal{D}_{i,i} =$

$\sum_{j=1}^{n} \mathcal{A}_{ij}$. The discrete Laplacian $\mathcal{L} := \mathcal{D} - \mathcal{A}$, and its right normalized version $\hat{\mathcal{L}} = I_{n \times n} - \mathcal{A}\mathcal{D}^{-1}$ are approximations to the Laplace-Beltrami operator Coifman & Lafon (2006).

A common approach to manifold learning in semi-supervised learning is based on the following two-step procedure (Belkin & Niyogi, 2008; 2006; Bengio et al., 2013; Hein et al., 2005; Hein & Maier, 2006): (I) approximate the Laplace-Beltrami operator over the manifold, by the discrete Laplacian matrix $\mathcal{L}$; and (II) compute a representation $\phi(x^{(i)})$ of each input token $x^{(i)}$ based on $\mathcal{L}$. Popular examples include the diffusion map Coifman & Lafon (2006) and Laplacian Eigenmaps.

The design of our first Transformer module $\mathsf{TF}_{rep}$ is motivated by the above two-step approach. We will further decompose $\mathsf{TF}_{rep}$ into two sub-modules: $\mathsf{TF}^{\mathcal{L}}$ (for forming a discrete Laplacican as in (I)) and $\mathsf{TF}^{\phi}$ (for computing the Eigenmap as in (II)). We prove by construction of $\mathsf{TF}_{rep} = \mathsf{TF}^{\phi} \circ \mathsf{TF}^{\mathcal{L}}$ that they *compute the Eigenmap $\phi$ of $\mathcal{L}_h$, given $x^{(1)}, \ldots, x^{(n)}$ as inputs*. In the following, we will consider two variants of standard attention, denoted by $\mathrm{Attn}_{linear}$ and $\mathrm{Attn}_{rbf}$. These two variants replace the standard softmax activation by the linear and RBF functions respectively; their detailed definitions, along with other implementation details, are presented in Appendix A.

**Transformer For Computing Discrete Laplacian.**
Let $X \in \mathbb{R}^{d \times n}$ denote the matrix whose $i^{th}$ column is the raw coordinates of the $i^{th}$ token $x^{(i)}$. Let $Z_0 \in \mathbb{R}^{(d+n) \times n}$ denote the concatenation of $X$ to $0_{n \times n}$, i.e. $Z_0^{\top} = [X^{\top}, 0_{n \times n}]$. This is equivalent to augmenting each $x^{(i)}$ with $0_d$. In Lemma 1 provided in Appendix A, we show that there exists a *simple Transformer* $\mathsf{TF}^{\mathcal{L}}$, which can compute the discrete Laplacian $\hat{\mathcal{L}}$, for any bandwidth $h$. Concretely, let $\mathsf{TF}^{\mathcal{L}}(Z_0) \in \mathbb{R}^{(d+n) \times n}$ denote the output of the 1-layer Transformer from Lemma 1. Then $[\mathsf{TF}^{\mathcal{L}}]_{d:d+n} = \mathcal{L}_h$. Notably, $\mathsf{TF}^{\mathcal{L}}$ has a *single-head*, a *single layer*, and all its parameter matrices are zero everywhere except for a diagonal block of (scaled) identity. Intuitively, such a simple construction exists because the $\mathrm{Attn}_{rbf}$ used in $\mathsf{TF}^{\mathcal{L}}$ exactly computes the $\exp(-\|x^{(i)} - x^{(j)}\|_2^2 /(2h))$ weights in $\mathcal{L}_h$. In practice, however, we use a multi-layer, multi-head Transformer for $\mathsf{TF}^{\mathcal{L}}$ for two reasons: first, multiple heads us to compute a *combination* of different Laplacians $\mathcal{L}_h$, each with a different RBF bandwidth $h$; second, the higher expressivity that comes with more layers enables the $\mathsf{TF}^{\mathcal{L}}$ to compute potentially more complex functions during end-to-end training.

**Transformer for computing Eigenmap.**
Let $\tilde{\mathcal{L}} = [\mathsf{TF}^{\mathcal{L}}(Z_0)]_{d:d+n} \in \mathbb{R}^{n \times n}$ denote the output of preceding Transformer stage. Given $\tilde{\mathcal{L}}$ as input, the next stage $\mathsf{TF}^{\phi}$ takes as input $Z_0^{\top} = [\tilde{\mathcal{L}}^{\top}, 0_{d \times d}, I_{d \times k}]$. The architecture is a modification of the construction in Cheng et al. (2024) for computing eigenvectors of graph Laplacians. In Lemma 2 provided in Appendix A.2, we show that there exists a Transformer $\mathsf{TF}^{\phi}$ that takes in $Z_0$ as defined above, and outputs the Laplacian Eigenmap for tokens $1...n$. Concretely, let the output of $\mathsf{TF}^{\phi}$ be

$$[\mathsf{TF}^{\phi}(Z_0)]^{\top} = [Z_1^{out\top}, Z_2^{out\top}, Z_3^{out\top}],$$

where $Z_1^{out} \in \mathbb{R}^{n \times n}$, $Z_2^{out} \in \mathbb{R}^{n \times n}$, $Z_3^{out} \in \mathbb{R}^{k \times n}$. Lemma 2 shows that $Z_3^{out} \approx [v_1...v_k]$, where $v_i$ is the $i^{th}$ smallest eigenvector of $\tilde{\mathcal{L}}$ in the input to $\mathsf{TF}^{\phi}$. *The $i^{th}$ column $Z_3^{out}$ constitutes an eigenmap of the $i^{th}$ token*. Henceforth, we use $\phi^{(i)}$ to denote the $i^{th}$ column of $Z_3^{out}$. The parameter configuration of $\mathsf{TF}^{\phi}$ is once again simple; its $W^Q, W^K, W^V$ matrices are all all diagonal. A key difference from $\mathsf{TF}^{\mathcal{L}}$ is that $\mathsf{TF}^{\phi}$ uses the *linear attention* $\mathrm{Attn}_{linear}$. This is important because linear attention is well-suited to implementing a power-method-like algorithm that efficiently computes the eigenvectors $v_1...v_k$. We defer details to the proof of Lemma 2 to Appendix A.2.

### 3.2 Transformer for In-context Supervised Learning

From the above discussion, for input covariate $x^{(i)}$, the feature-representation stage of our model yields associated features $\phi^{(i)}$, which depend on *all* covariates $x^{(1)}, \ldots, x^{(n)}$. We now discuss how to utilize these features in the subsequent ICL stage with categorical observations.

Our derivation is based on the assumption that $f(\phi)$ resides in a reproducing kernel Hilbert space (RKHS) Schölkopf & Smola (2002), but the setup extends to softmax-based attention kernels as well Wang et al. (2025); Cheng et al. (2024). From the RKHS perspective, let $f(\phi) = A\psi(\phi)$, with $\psi(\phi)$ a *fixed* mapping of features $\phi$ to a Hilbert space, and the parameters acting in that space are associated with matrix $A$.

Assuming we have $m$ labeled samples in a given contextual dataset, indexed $i = 1, \ldots, m$, the cross-entropy cost function for inferring the parameters $A \in \mathbb{R}^{d' \times m}$ may be expressed as

$$\mathcal{L}(A) = -\frac{1}{m} \sum_{i=1}^{m} \log \left[ \frac{\exp[w_{y_i}^T (A\psi(\phi^{(i)}))]}{\sum_{c=1}^{C} \exp[w_c^T (A\psi(\phi^{(i)}))]} \right] \tag{5}$$

We emphasize that this loss is minimized anew for each contextual dataset, and it will be implemented in the Transformer forward pass, for Transformer $\mathsf{TF}_{sup}$. Performing gradient descent (GD) wrt parameters $A$, and using $\kappa(\phi^{(i)}, \phi^{(j)}) = \psi(\phi^{(i)})^\top \psi(\phi^{(j)})$ one yields the functional update rule for $i = 1, \ldots, n$ (all samples) (see Appendix B)

$$f_{\ell+1}^{(i)} = f_\ell^{(i)} + \Delta f_\ell^{(i)}, \qquad \Delta f_\ell^{(i)} = \sum_{j=1}^{m} \alpha \left[ w_{y^{(j)}} - \mathbb{E}(w | f_\ell^{(j)}) \right] \kappa(\phi^{(i)}, \phi^{(j)}) \tag{6}$$

$$\mathbb{E}(w | f_\ell^{(j)}) = \sum_{c=1}^{C} w_c \left[ \frac{\exp[w_c^T f_\ell^{(j)}]}{\sum_{c'=1}^{C} \exp[w_{c'}^T f_\ell^{(j)}]} \right] \approx \mathrm{MLP}_\gamma(f_\ell^{(j)}) \tag{7}$$

with $\alpha$ the GD learning rate. Importantly, in (6) the labeled data are employed within the sum over $j = 1, \ldots, m$, and the relationship to the unlabeled data for $i = m + 1, \ldots, n$ is handled by the kernel $\kappa(\phi^{(i)}, \phi^{(j)})$. Hence, we see that the representation-learning stage of the Transformer, discussed above, has the task of designing features $\phi^{(i)}$ such that the kernel $\kappa(\phi^{(i)}, \phi^{(j)})$ captures the appropriate interrelationship between the data samples for predicting associated labels.

The form of the update in (6) is the same as that considered in previous Transformer ICL research von Oswald et al. (2023); Ahn et al. (2023); Zhang et al. (2023); Schlag et al. (2021); Cheng et al. (2024); Wang et al. (2025), and it naturally aligns with a self-attention layer (which considered real-valued observations).

Once the vector $f_\ell^{(i)}$ is updated to $f_{\ell+1}^{(i)} = f_\ell^{(i)} + \Delta f_\ell^{(i)}$, the expectation in (7) must be updated. Expectation (7) may be viewed as a nonlinear function of (now) $f_{\ell+1}^{(i)}$. Therefore, the attention layer is followed by an MLP layer, represented in (7) by $\mathrm{MLP}_\gamma(f_{\ell+1}^{(i)})$, with parameters $\gamma$. The MLP parameters $\gamma$ are learned along with all other parameters of the Transformer. Details of the MLP layer and the other elements of this ICL Transformer are provided in Appendix B.

## 4 EXPERIMENTS

This section describes the benchmarks we devised to study the semi-supervised, in-context learning capabilities of our end-to-end Transformer. After presenting the baseline comparative benchmark models, and the data-generation pipelines, we discuss the evaluations we report. For all experiments, we vary the percentage of labeled data among the fixed $n = 100$ total samples. Unless stated otherwise, all results that employ ICL for classification use a one layer ICL Transformer (as in Section 3.2) and utilize the RBF kernel as the core component of the GD ICL head. Further implementation details required for exact replication are collected in Appendices A and B.

Our evaluation focuses on three key aspects: (1) the ability of the Transformer to learn meaningful representations from unlabeled data, (2) the effectiveness of the end-to-end approach compared to traditional methods, (3) the generalization capabilities across different manifolds and data distributions, and (4) mechanistic understanding of in-context semi-supervised learning (IC-SSL) on real-world data through an ImageNet100 experiment.

### 4.1 MODELS UNDER EVALUATION:

We consider the following models under the naming notation of *[Data+Model]*, reflective of the form of input to the classifier (Data), and the model used to classify the unlabled samples (Model).

**ORIG+E2E-ICL (OUR MODEL)** The raw *original* coordinate matrix is fed directly into our **end-to-end Transformer**. See Section 3 and Appendix A for a full architectural breakdown. Crucially, all three constituent modules—the Laplacian predictor, the eigenvector extractor, and the in-context GD head—are trained *jointly* minimizing their combined loss.

**EIG+ICL**  The ICL head uses *ground-truth* bottom-$k$ Laplacian eigenvectors as input, isolating the performance contribution of representation learning.

**EIG+LR**  Standard $\ell_2$-regularized logistic regression on *ground-truth* eigenvector features.

**ORIG+RBF-LR**  Kernel logistic regression with Gaussian RBF kernel on the *original* coordinates.

**ORIG+ICL**  The in-context learning head is applied directly on the *original* coordinates, without Laplacian eigenvector features.

### 4.2  IMAGENET100: SEMI-SUPERVISED IC-SSL ON REAL DATA

**Experimental Design.**  We evaluate on ImageNet100, a 100-class subset of ILSVRC-2012 Russakovsky et al. (2015) commonly used for efficient representation learning Caron et al. (2018); Chen et al. (2020). Here, the "dataset size" denotes the number of in-context datasets (tasks). Each dataset has 200 batches, each batch sampling two random classes (50 images per class, 100 total) *with replacement*, and VGG-29 features Simonyan & Zisserman (2014) are used as inputs. The batch size is fixed at 200, with $3\%$ labeled samples (a $39\%$ variant is reported in Appendix F.4). For comparison, we use a scale-matched Transformer baseline (2-layer encoder, hidden dimension 128, $\sim$1.5M parameters), where class labels are projected into the same embedding space as the inputs and unlabeled samples are assigned a special `unknown` token, following the MetaICL formulation Min et al. (2022) to ensure fairness. Ground-truth **EIG** embeddings are obtained via Laplacian eigendecomposition of the affinity graph as a spectral reference.

We compare three systems: (i) our *constructed* two-stage model, (ii) a scale-matched 2-layer *baseline Transformer* trained end-to-end on the same episodes, and (iii) an *ICL head on raw VGG features* (Orig+ICL). We report standard ICL accuracy and a simple *separation score* Caron et al. (2018); Oord et al. (2018); Padhan & Sahu (2024) (intra-class similarity minus inter-class similarity), which serves as a coarse proxy for representation quality (Figure 3).

**Accuracy and separation.** Our model attains higher accuracy than the baseline Transformer, with the largest gaps appearing at small training-set sizes; both models outperform EIG+ICL, indicating that *in-context representation learning* is crucial and that raw VGG features have poor separation. Separation and accuracy are correlated across data regimes: our model exhibits higher separation even with little data, and the accuracy advantage mirrors this difference (Figure 3).

**Phase transition in the baseline Transformer.** The baseline Transformer performs poorly when the training set is smaller than $\approx 1000$ and undergoes a sharp *phase transition* around $\sim 1200$, after which accuracy rises quickly. Strikingly, this jump aligns *exactly* with an abrupt increase in separation score, linking the performance transition to a change in representational structure (Figure 3).

**Representational alignment.** To connect these trends to geometry, we use mutual $k$NN (mNN) alignment: for each sample, take its top-$k$ neighbors in two embeddings and compute the fraction of overlap between the two neighbor sets; the mNN score is the dataset-average (higher = more similar neighborhood structure) Tsuchiya et al. (2023). We compare three embeddings—ours, the baseline Transformer, and *EIG* (top-4 Laplacian eigenvectors from the affinity graph). Alignment rises with training data. Our model starts highly aligned to EIG (spectral inductive bias), while the baseline progressively aligns with both; with enough data, *baseline–ours* can exceed *ours–EIG*, and *baseline–EIG* improves but generally trails *ours–EIG* at moderate scales. Additional metrics (Cycle, LCS) show the same pattern; by 5000 episodes all three pairs exhibit high alignment (Table 1).

Table 1: Alignment metrics at labeled ratio $3\%$ and dataset size $=$ 5000, comparing three embeddings: (**Our model**), (**Transformer baseline**), and (**EIG**). We report three alignment metrics (higher is better) introduced in the Platonic Representation Hypothesis Tsuchiya et al. (2023).

| Source–Target Pair | mNN ↑ | Cycle ↑ | LCS ↑ |
|---|---|---|---|
| Our model – Transformer baseline | 0.302 | 0.289 | 0.193 |
| EIG – Our model | 0.304 | 0.295 | 0.195 |
| EIG – Transformer baseline | 0.269 | 0.262 | 0.177 |

Figure 3 and Table 1 show our two-stage model is competitive on ImageNet-based artificial IC-SSL tasks, using real image features, provides mechanistic insight via a phase transition in the baseline, and serves as a *reference model* for how end-to-end Transformers acquire PE-like structure.

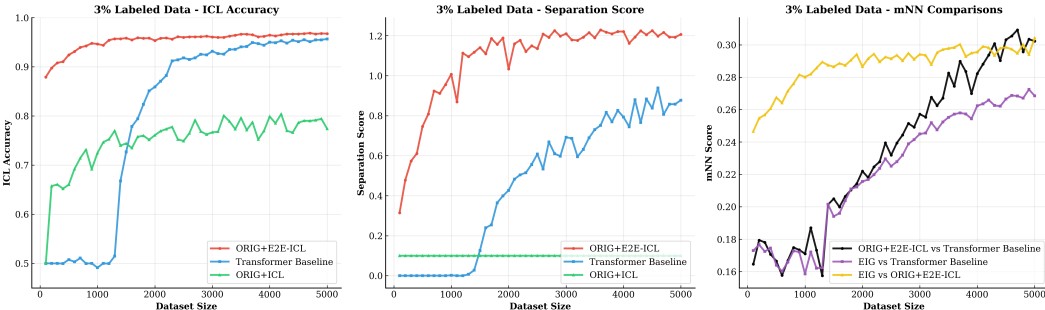

Figure 3: In-context learning on ImageNet100 with 3% labels; "dataset size" = number of constructed in-context tasks. Columns: left—ICL accuracy; middle—separation (intra- minus inter-class similarity); right—mNN neighborhood-overlap similarity. Methods: **Orig+E2E-ICL** (red), **Transformer baseline** (blue), **Orig+ICL** (green; VGG-29 features).

**Data efficiency and spectral perspective.** In a representative low-label regime (3% labels, $N$=5000), our model improves both accuracy and separation more rapidly than the baseline Transformer, reflecting faster emergence of intra-class compactness and inter-class separation (Figure 3). While both models eventually reach comparable performance at larger data scales, ours does so with substantially fewer samples. From a spectral viewpoint, both representations appear to approximate the same underlying geometry: alignment to the "EIG" (top Laplacian eigenvectors) strengthens with more data, with our model starting closer in low-data settings due to its inductive bias, and the baseline progressively converging toward both our model and the spectral reference (Table 1).

### 4.3 FAMILIES OF MANIFOLDS AND THEIR GEOMETRY

Let $(\mathcal{M}, g)$ denote a smooth, connected Riemannian manifold with intrinsic distance $d_{\mathcal{M}}$. We use five such manifolds, visualized in Figure 2. Each manifold admits a closed-form geodesic and a parametric map $\Phi \colon U \subset \mathbb{R}^m \to \mathbb{R}^3$. For example, for the sphere $\mathbb{S}^2$, $\Phi_{\mathbb{S}^2}(\theta, \phi) = (\sin\theta\cos\phi, \ \sin\theta\sin\phi, \ \cos\theta)$, and its geodesic is $d_{\mathbb{S}^2}(p, q) = \arccos(\langle p, q \rangle)$. Details of their parametric maps and geodesic derivations are in Appendix E. Many of these manifolds have had important applications in scientific machine learning (De Bortoli et al., 2022; Jing et al., 2022).

In our experiments, we sample $n = 100$ points uniformly from the manifold in $\mathbb{R}^3$, followed by random rotations + translations (see Appendix E) to generate a family of perturbed manifolds. For each set, we select a random center point and assign binary labels based on geodesic distance: points within a certain manifold-distance form the positive class, while others form the negative class.

#### 4.3.1 RESULTS ON INDIVIDUAL MANIFOLDS

**In-distribution performance.** For each elementary manifold we train and evaluate every model on data drawn from that same manifold—e.g., the "cylinder" curves in Figure 4a are obtained by training exclusively on cylinder samples and measuring accuracy on a disjoint cylinder test split. The figure shows the mean ± one standard deviation over three random initialisations, where each run uses an 90 %/10 % train–test split, identical label budgets, and fresh random draws of the unlabeled context points. The same protocol applied to the sphere, cone, torus and Swiss-roll manifolds yields curves with the same qualitative ordering; those results are deferred to Appendix F.3.

For the cylinder our ORIG+E2E-ICL model dominates all baselines across the entire label-budget spectrum, surpassing the strongest competitor (EIG+ICL) by roughly 5–7 percentage points and reaching $\sim 90\,\%$ accuracy once $\geq 25\%$ of the points are labelled. The advantage is pronounced in the low-label regime ($3\% \leq \rho \leq 15\%$), confirming that end-to-end contextual representation learning is particularly valuable when supervision is scarce. Notably, direct classification on raw coordinates (ORIG+RBF-LR) performs markedly worse than any manifold-aware alternative, highlighting the benefit of exploiting geometric structure rather than relying on purely supervised fits.

**Kernel choice in the ICL head.** Replacing the RBF kernel with a linear kernel changes the performance and occasionally the ordering, but the dominance of the end-to-end approach remains; detailed ablations appear in Appendix F.2.

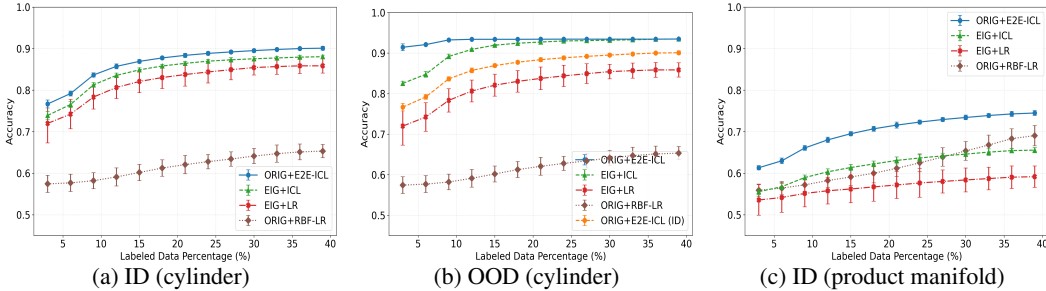

|  |  |  |
|---|---|---|
| (a) ID (cylinder) | (b) OOD (cylinder) | (c) ID (product manifold) |

Figure 4: Accuracy vs. labeled-sample ratio on manifold benchmarks. (a) In-distribution: train/test on cylinder. (b) OOD (cylinder): blue ORIG+E2E-ICL and green ORIG+ICL are trained on {sphere, cone, torus, Swiss-roll}, tested on cylinder; orange is the ID reference (train/test on cylinder). (c) In-distribution (product manifold): training and testing on the high-dimensional product manifold ($\mathbb{S}^2 \times C \times \text{Cone}_\alpha \times \text{SR} \times \mathbb{T}^2$).

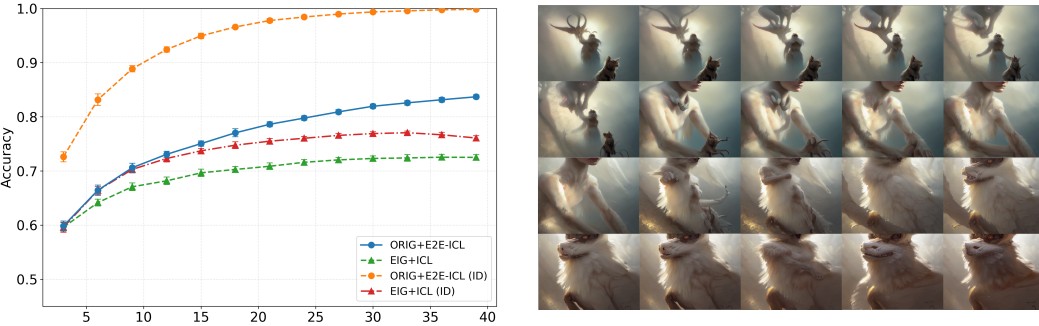

Figure 5: Results on image manifolds. Left: zero-shot test accuracy, where blue ORIG+E2E-ICL and green EIG+ICL are trained only on four synthetic manifolds {sphere, cone, Swiss-roll, cylinder} and tested on image manifolds; red/orange denote ID runs (trained and tested on images). Baselines EIG+LR/ORIG+RBF-LR are omitted for large underperformance. Right: an example manifold obtained by sub-sampling every 5th of $n = 100$ SLERP frames between two random latents; top two rows are label 1, bottom two rows are label 0, illustrating the smooth geodesic manifold structure.

**Out-of-distribution (OOD) evaluation.** To test generalizability, we train on four source manifolds— sphere, cone, torus, and Swiss-roll— and test on the held-out cylinder. Figure 4b contrasts this OOD setting with an in-distribution (ID) baseline trained and tested on cylinder (Figure 4a). Across label budgets, ORIG+E2E-ICL matches or exceeds its ID performance and is 3–5 percentage points higher in this case; other OOD trials (see Appendix F.3) generally approach but do not surpass their ID counterparts. These results indicate that representations learned from diverse geometries transfer effectively to unseen manifolds. The logistic-regression baselines (EIG+LR, ORIG+RBF-LR) are refit on cylinder at test time and thus act as ID references rather than true OOD comparators; EIG+ICL underperforms the end-to-end model.

### 4.3.2 PRODUCT MANIFOLD RESULTS

The manifolds described previously live in $\mathbb{R}^3$ and possess relatively simple decision boundaries. To further evaluate the representation learner, we form Cartesian *product manifolds* $\mathcal{P}_K = \mathcal{M}_1 \times \cdots \times \mathcal{M}_K$, where each factor $\mathcal{M}_k$ is drawn from the five families of manifolds we individually tested against. The intrinsic dimension therefore grows additively with $K$, and likewise the complexity of the geodesic ball that defines the label boundary increases as well. As a consequence of our Theorem 1 (see Appendix E.2.1), the geodesic distance on $\mathcal{P}_K$ remains available in closed form, allowing us to generate exact labels *in the exact same way as before* even in the presence of the random scale, rotation, and permutation applied independently to each factor. Details on these transformations, the decision thresholds, and the construction are provided in Appendix E.

Figure 4c reports accuracy on a representative 5-factor product manifold of all of our individual manifold families. As expected, all the methods degrade as geometry becomes more intricate, yet the ORIG+E2E-ICL Transformer retains an $8-10\%$ margin over the strongest baseline (EIG+ICL) across the whole label spectrum. These results support that the end-to-end learner continues to extract useful Laplacian-based features and to propagate labels effectively even when the decision boundary is more complex and in a space whose dimensionality exceeds that of any single chart.

## 4.4 IMAGE MANIFOLD EXPERIMENTS

To test whether the representation-learning routine transfers to higher dimensional, natural-looking data, we build an *image manifold* with Stable Diffusion v1.5 Podell et al. (2023). Concretely, we draw two latent noise vectors, treat them as endpoints, and generate a 100-step spherical linear interpolation in latent space. Each interpolated latent is decoded—without classifier guidance—into a $500 \times 500$ RGB image. Repeating this over 500 prompt seeds yields 500 diverse sub-manifolds of images (each composed of 100 images). The intrinsic geometry of each sub-manifold is inherited from the latent interpolation; full generation and labeling details appear in Appendix E.3.

The results demonstrate exceptional zero-shot OOD transfer capabilities: despite having been trained exclusively on simple synthetic manifolds from before, with no prior exposure to the image data, our model achieves roughly 77% accuracy using only 15% labeled examples; and remarkably sustains performance above 62% accuracy even at extremely spare labeling (3/100 labeled images). This consistently surpasses baselines methods by margins of 10-20 percentages points across nearly the entire labeled data regime. For reference, our model trained directly on the image manifold achieves nearly 100% accuracy, far surpassing the other methods.

Perhaps more striking than the headline accuracies is the data efficiency: three labels guide the Transformer to a correctness level that baselines cannot match even after seeing 10x more labels. This suggests the network is implementing an efficient modality-agnostic algorithm for constructing a geometry-aware representation from raw coordinates at inference time. The sparse supervision is propagated through this learned geometry, leading to impressive label economy.

## 5 LIMITATIONS AND CONCLUSION

We introduced semi-supervised in-context learning with a single end-to-end Transformer that first builds a geometry-aware representation from all covariates and then predicts labels in-context—without parameter updates. Across synthetic manifolds, product manifolds, and image manifolds, our model consistently outperforms strong baselines and shows robust out-of-distribution transfer. We use our Transformer model as a reference for studying the IC-SSL capabilities of a standard Transformer. We observe correlations in the representations learned by our model and by the standard Transformer, suggesting that representation learning may be an important direction in understanding the Transformer's IC-SSL capabilities. While the construction is applicable in principle to high-dimensional data, our experiments use relatively small episode-style datasets, reflecting the novelty of the IC-SSL setting and the absence of established benchmarks.

## 6 ACKNOWLEDGEMENTS

We thank Zeyu Michael Li for his crucial contributions to the image manifold experiments. These contributions warranted authorship; however, he was inadvertently omitted from the author list at the time of submission and could not be added retroactively under ICLR policy. Zeyu is listed as a co-author on the arXiv version of this paper.

## 7 Ethics Statement

This work uses (i) synthetic manifolds and product manifolds we generate procedurally, (ii) images produced by Stable Diffusion v1.5 without human subjects, and (iii) publicly available ImageNet100 features under their licenses. No personally identifiable information or sensitive attributes are collected or inferred. Potential risks are limited to standard concerns in semi-supervised learning (e.g., amplification of dataset bias); we discourage deployment without bias and robustness assessment on target data. No other foreseeable dual-use concerns were identified.

## 8 Reproducibility Statement

An anonymized code archive, here, is provided with scripts and configs to regenerate figures and tables. Data generation pipelines are specified for: classic manifolds and product manifolds (Appendix E, Theorem/derivation in Appendix E.2.1), the Stable-Diffusion image manifolds (Appendix E.3), and the ImageNet100 episodic setup (Section 4.2). Model architectures, loss weighting, optimization settings, and training schedules are listed in Section 3 and Appendix D; derivations and construction details for the representation module and ICL head appear in Appendices A and B. Baselines and their hyperparameters are documented in Section F.7 and Appendix D. We fix random seeds and report mean $\pm$ sd over three runs; the repository provides code needed to reproduce our work and further details.

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

# APPENDIX

## A  Transformer for Representation Learning (Lemmas 1 and 2)

Given samples $z^{(1)}, \ldots, z^{(n)} \in \mathbb{R}^d$, let $Z \in \mathbb{R}^{d \times n}$ denote the matrix whose $i^{th}$ column is $z^{(i)}$. Let $\kappa(U, V)$ denote a non-linear transformation. We define the $\kappa$-attention module as

$$\mathrm{Attn}_\kappa(Z; W^V, W^Q, W_k) = W^V Z \kappa\big(W^Q Z, W_k Z\big) \tag{8}$$

In standard softmax-activated attention, $[\kappa(U, V)]_{i,j} := \exp\left(U^{(i)^\top} V^{(j)}\right) / \sum_{k=1}^n \exp\left(U^{(i)^\top} V^{(k)}\right)$. For the purpose of this section, we will consider two other types of activations:

$$[\kappa_{\mathtt{linear}}(U, V)]_{i,j} := U^{(i)^\top} V^{(j)}$$

$$[\kappa_{\mathtt{rbf}}(U, V)]_{i,j} := \frac{\exp\left(-\|U^{(i)} - V^{(j)}\|_2^2\right)}{\sum_{k=1}^n \exp\left(-\|U^{(k)} - V^{(j)}\|_2^2\right)}, \tag{9}$$

where $U^{(i)}$ denotes the $i^{th}$ column of $U$. Throughout this section, we will consider the Transformer defined by the following update:

$$Z_{\ell+1} = \big(I + W_\ell^S\big) Z_\ell + \sum_{h=1}^H \mathrm{Attn}_\kappa(Z_\ell; W_{\ell,h}^V, W_{\ell,h}^Q, W_{\ell,h}^K), \tag{10}$$

where $W_\ell^S$ is a linear transformation, $\ell$ is the layer number, $H$ is the total number of heads, and $\{W_{\ell,h}^V, W_{\ell,h}^Q, W_{\ell,h}^K\}$ are the value, query and key matrices, respectively.

### A.1  Transformer for forming normalized graph Laplacian (Lemma 1)

The Transformer that we use for learning a Laplacian-based representation is based on the one proposed in Section 14.2 of Cheng et al. (2025) with one important modification – replacing the linear attention by RBF attention – enables the same architecture to efficiently compute the Laplacian of the nearest-neighbors graph in-context. See (9) above for definitions of $\mathrm{Attn}_{\mathtt{rbf}}$.

Let $x^{(1)}, \ldots, x^{(n)}$ denote the coordinate representation of token $i$. Let $X \in \mathbb{R}^{d \times n}$ denote the matrix whose $i^{th}$ column is given by $X^{(i)} = x^{(i)}$. We will begin by forming the *augmented token* $z^{(i)} = [x^{(i)}; 0_n] \in \mathbb{R}^{d+n}$. Let $Z_0 \in \mathbb{R}^{(d+n) \times n}$ denote the matrix whose columns are $z^{(i)}$. The input to the Transformer is thus $Z_0^\top = [X^\top, 0_{n \times n}]$. We have used $0_n$ and $0_{n \times n}$ to denote all-zeros vectors and matrices, of dimensions indicated by the subscripts. In practice, we will also learn an additional initialization parameter matrix $M \in \mathbb{R}^{n \times n}$, so that

$$Z_0^\top = [X^\top, M].$$

Adding an initialization $M$ ensures that the architecture is identical to the Eigenmap Transformer in the subsequent section (except for choice of nonlinear activation), at no loss of expressivity.

Let $Z_\ell \in \mathbb{R}^{d+n}$ denote the output of the $\ell^{th}$ Transformer layer, as defined in (10), with $\mathrm{Attn}_{\mathtt{rbf}}$. We impose the following constraints on the parameter matrices of the Transformer. To reduce notation, we focus on the single-head setting; the case of multiple heads simply repeats the following construction in parallel:

$$W_\ell^V = \begin{bmatrix} a_\ell I_{d \times d} & 0_{d \times n} \\ 0_{n \times d} & A_\ell \end{bmatrix}, \; W_\ell^Q = \begin{bmatrix} b_\ell I_{d \times d} & 0_{d \times n} \\ 0_{n \times d} & B_\ell \end{bmatrix}, \; W_\ell^K = \begin{bmatrix} c_\ell I_{d \times d} & 0_{d \times n} \\ 0_{n \times d} & C_\ell \end{bmatrix},$$

$$W_\ell^S = \begin{bmatrix} s_\ell I_{d \times d} & 0_{d \times n} \\ 0_{n \times d} & S_\ell \end{bmatrix}, \tag{11}$$

where $A_\ell, B_\ell, C_\ell, S_\ell \in \mathbb{R}^{n \times n}$.

The total set of Transformer parameters is

$$\{M\} \cup \{s_\ell, S_\ell, a_\ell, A_\ell, b_\ell, B_\ell, c_\ell, C_\ell\}_{\ell=1,\ldots,L}.$$

**Remark 1.** *In our construction of Lemma 1, the matrices $A_\ell = a'_\ell I_{n \times n}, B_\ell = b'_\ell I_{n \times n}, C_\ell = c'_\ell I_{n \times n}, S_\ell = s'_\ell I_{n \times n}$, where $a'_\ell, b'_\ell, c'_\ell, s'_\ell$ are scalar multiples of the identity matrix. In our implementation, $a'_\ell, b'_\ell, c'_\ell, s'_\ell$ are learnable scalar parameters, and thus the parameter count does not scale with $n$.*

We show in the following lemma that a 1-layer Transformer with $h = 1$ computes the (RBF) Laplacian with bandwidth $\sigma^2$. For ease of reference, we redefine several matrices of interest below:

1. The $\sigma^2$-bandwidth RBF adjacency matrix $\mathcal{A} \in \mathbb{R}^{n \times n}$ is defined by

$$\mathcal{A}_{ij} = \exp\left(-\frac{1}{\sigma^2}\|x^{(i)} - x^{(j)}\|_2^2\right). \tag{12}$$

2. The diagonal matrix $\mathcal{D} \in \mathbb{R}^{n \times n}$ is defined with diagonal elements $D_{jj} := \sum_{k=1}^n \exp\left(-\frac{1}{\sigma^2}\|x^{(k)} - x^{(j)}\|_2^2\right)$.

3. The random walk matrix is defined as $\mathcal{A}\mathcal{D}^{-1} \ni \mathbb{R}^{n \times n}$. We verify that

$$\left[\mathcal{A}\mathcal{D}^{-1}\right]_{ij} = \frac{\exp\left(-\frac{1}{\sigma^2}\|x^{(i)} - x^{(j)}\|_2^2\right)}{\sum_k \exp\left(-\frac{1}{\sigma^2}\|x^{(k)} - x^{(j)}\|_2^2\right)}. \tag{13}$$

4. The Laplacian $\mathcal{L}$, symmetric-normalized Laplacian $\bar{\mathcal{L}}$ and right-normalized Laplacian $\hat{\mathcal{L}}$ are defined as follows:

$$\mathcal{L} = \mathcal{D} - \mathcal{A}, \qquad \bar{\mathcal{L}} = I_{n \times n} - \mathcal{D}^{-1/2}\mathcal{A}\mathcal{D}^{-1/2}, \qquad \hat{\mathcal{L}} = I_{n \times n} - \mathcal{A}\mathcal{D}^{-1} \tag{14}$$

**Lemma 1** (Construction of Transformer for Computing Laplacian). *Let $\hat{\mathcal{L}} \in \mathbb{R}^{n \times n}$ denote the right-normalized Laplacian with bandwidth $\sigma^2$ as defined in (14) above. Consider the single-head Transformer, parameterized as in (11), with the following parameters: $s_\ell = 0, S_\ell = 0_{n \times n}, a_\ell = 0, b_\ell = c_\ell = 1/\sigma, A_\ell = -I_{n \times n}, B_\ell = C_\ell = 0_{n \times n}$. Let $[Z_1]_{d:n+1} \in \mathbb{R}^{n \times n}$ denote the last $n$ rows of $Z_1$. Let $M = I_{d \times d}$. Then $[Z_1]_{d:d+n} = \hat{\mathcal{L}}$.*

*Proof of Lemma 1.* For convenience, let $Z_\ell^X \in \mathbb{R}^{d \times n}$ denote the first $d$ rows of $Z_\ell$ and $Z_\ell^\phi$ denote the last $n$ rows of $Z_\ell$. Let $\kappa_{\ell,h} \in \mathbb{R}^{n \times n}$ denote the self-similarity matrix. We begin by verifying that for $\ell = 0$, the self-similarity matrix simplifies to

$$
\begin{aligned}
\left[\kappa_{\mathrm{rbf}}\left(W_\ell^K Z_\ell, W_\ell^Q Z_\ell\right)\right]_{ij} &= \frac{\exp\left(-\left\|[W_\ell^Q Z_\ell]^{(i)} - [W_\ell^K Z_\ell]^{(j)}\right\|_2^2\right)}{\sum_k \exp\left(-\left\|[W_\ell^Q Z_\ell]^{(k)} - [W_\ell^K Z_\ell]^{(j)}\right\|_2^2\right)} \\
&= \frac{\exp\left(-b_\ell c_\ell\left\|x^{(i)} - x^{(j)}\right\|_2^2\right)}{\sum_k \exp\left(-b_\ell c_\ell\left\|x^{(k)} - x^{(j)}\right\|_2^2\right)} \\
&= \frac{\exp\left(-\frac{1}{\sigma^2}\left\|x^{(i)} - x^{(j)}\right\|_2^2\right)}{\sum_k \exp\left(-\frac{1}{\sigma^2}\left\|x^{(k)} - x^{(j)}\right\|_2^2\right)}.
\end{aligned}
$$

From the above, we verify that $\kappa_{\mathrm{rbf}}\left(W_\ell^K Z_\ell, W_\ell^Q Z_\ell\right) = \mathcal{A}\mathcal{D}^{-1}$, which is the random walk matrix, defined in (12) and (13). By definition of $M$ and $Z_0$, we have

$$W_0^V Z_0 = \begin{bmatrix} 0_{d \times n} \\ I_{n \times n} \end{bmatrix}.$$

Consequently, for $\ell = 0$,

$$
\begin{aligned}
\mathrm{Attn}_{\mathrm{rbf}}(Z_\ell) &= W_\ell^V Z_\ell \kappa_{\mathrm{rbf}}\left(W_\ell^K Z_\ell, W_\ell^Q Z_\ell\right) \\
&= \begin{bmatrix} 0_{d \times n} \\ -I_{n \times n} \end{bmatrix} \mathcal{A}\mathcal{D}^{-1} = \begin{bmatrix} 0_{d \times n} \\ -\mathcal{A}\mathcal{D}^{-1} \end{bmatrix}
\end{aligned}
$$

which is exactly equal to $\mathcal{A}_{ij}$, where $\mathcal{A}$ is the adjacency matrix defined in Section 3.1. Finally, by definition of $s_\ell = 0$ and $S_\ell = I_{n \times n}$, we have

$$Z_1 = Z_0 - \begin{bmatrix} 0_{d \times n} \\ \mathcal{A}\mathcal{D}^{-1} \end{bmatrix} = \begin{bmatrix} X \\ I_{n \times n} \end{bmatrix} - \begin{bmatrix} 0_{d \times n} \\ \mathcal{A}\mathcal{D}^{-1} \end{bmatrix} = \begin{bmatrix} X \\ \hat{\mathcal{L}} \end{bmatrix},$$

where $\hat{\mathcal{L}}$ is as defined in (14). $\qquad\square$

### A.2 Transformer for Computing Eigenmap (Lemma 2)

As with the preceding section, the Transformer that we use for computing the Eigenmap is based on the Transformer from in Lemma 9 of Cheng et al. (2025) with one important modification. Here, we retain the use of $\mathrm{Attn_{linear}}$ as our goal is to implement a block-power-method algorithm for computing the Eigenvectors of a matrix. See (9) above for definitions of $\mathrm{Attn_{linear}}$.

Let $\psi^{(1)}, \ldots, \psi^{(n)} \in \mathbb{R}^n$ denote some pre-computed representation of $i$. Let $\Psi := [\psi^{(1)}, \ldots, \psi^{(n)}] \in \mathbb{R}^{n \times n}$ denote the matrix whose columns contain representations of each token. We are primarily interested in the setting when $\Psi$ is the Laplacian matrix $\hat{\mathcal{L}}$, output by the Transformer constructed in Lemma 1; however, the construction we present below applies to a general $\Psi$. This generalization is practically important, as it enables the Transformer to learn to compute embeddings not based on the RBF Laplacian.

Let $Z_0 \in \mathbb{R}^{(n+k) \times n}$ denote the matrix whose columns are $z^{(i)}$. The input to the Transformer is thus $Z_0^\top = [\Psi^\top, 0_{n \times k}]$. In practice, we will also learn an additional initialization matrix $M \in \mathbb{R}^{k \times n}$, so that

$$Z_0^\top = [\Psi^\top, 0_{n \times k} + M^\top].$$

On a high level, the Transformer will perform a block-power-method algorithm using the $\Psi$ matrix to compute the bottom $k$ eigenvectors of $Z_0$, and $M$ enables the Transformer to learn a good initial guess of the $k$ eigenvectors.

Let $Z_\ell \in \mathbb{R}^{d+n}$ denote the output of the $\ell^{th}$ Transformer layer, as defined in (10) (single head), but with $\mathrm{Attn_{linear}}$, and an additional normalization between layers:

$$\begin{aligned} Z_{\ell+1} &= (I + W_\ell^S)Z_\ell + \mathrm{Attn_{linear}}(Z_\ell; W_\ell^V, W_\ell^Q, W_\ell^K), \\ Z_{\ell+1} &= \mathtt{normalize}(Z_{\ell+1}), \end{aligned} \tag{15}$$

where $\mathtt{normalize}(Z)$ applies row-wise normalization to $Z$. We impose the following constraints on the parameter matrices of the Transformer.

$$W_\ell^V = \begin{bmatrix} a_\ell I_{n \times n} & 0_{n \times k} \\ 0_{k \times n} & A_\ell \end{bmatrix}, W_\ell^Q = \begin{bmatrix} b_\ell I_{n \times n} & 0_{n \times k} \\ 0_{k \times n} & B_\ell \end{bmatrix}, W_\ell^K = \begin{bmatrix} c_\ell I_{n \times n} & 0_{n \times k} \\ 0_{k \times n} & C_\ell \end{bmatrix},$$

$$W_\ell^S = \begin{bmatrix} s_\ell I_{n \times n} & 0_{n \times k} \\ 0_{k \times n} & S_\ell \end{bmatrix}, \tag{16}$$

where $A_\ell, B_\ell, C_\ell, S_\ell \in \mathbb{R}^{k \times k}$. Thus the total set of Transformer parameters is

$$\{M\} \cup \{s_\ell, S_\ell, a_\ell, A_\ell, b_\ell, B_\ell, c_\ell, C_\ell\}_{\ell=1,\ldots,L}.$$

**Remark 2.** *In our construction of Lemma 2, the matrices $A_\ell, B_\ell, C_\ell, S_\ell$ are $k \times k$ parameter matrices which do not scale with $n$.*

**Lemma 2** (Construction of Transformer for computing Eigenmap.). *Let $\Psi \in \mathbb{R}^{n \times n}$ be some feature map. Let $\mu \in \mathbb{R}^+$ denote a constant that satisfies $\mu I \succ \Psi^\top \Psi$. Then there exists a Transformer of the form* (15), *whose parameter matrices satisfy the constraints of* (16), *which implements the block-power-iteration algorithm for $\mu I - \Psi^\top \Psi$. Consequently, let $Z_L \in \mathbb{R}^{(n+k) \times n}$ denote the output of a $L$-layer Transformer, let $\Phi \in \mathbb{R}^{k \times n}$ denote the last $k$ rows of $Z_L$, and let $\Phi_j$ denote the $j^{th}$ row of $\Phi$. Then*

$$\Phi_j \approx v_j,$$

*where $v_j$ is the $j^{th}$ smallest eigenvector of $\Psi^\top \Psi$. When $\Psi$ is the Laplacian $\hat{\mathcal{L}}$, the bottom $k$ eigenvectors of $\Psi^\top \Psi$ are the same as the bottom $k$ right-singular vectors of $\hat{\mathcal{L}}$. When $\hat{\mathcal{L}}$ has constant degree (i.e. $\mathcal{D} \propto I_{n \times n}$), these are also equal to the bottom $k$ eigenvectors of the unnormalized Laplacian $\mathcal{L}$ and the symmetric-normalized Laplacian $\bar{\mathcal{L}}$.*

**Remark 3.** *Before stating the proof, we make a number of remarks on how the above result relates to Section 3.1.*

1. *Notice that the Laplacian Transformer $\mathsf{TF}^{\mathcal{L}}$ from Lemma 1 has input/output dimension $d + n$ and the Eigenmap Transformer $\mathsf{TF}^{\Psi}$ from Lemma 2 has input/output dimension $n + k$. For consistency, we pad each Transformer accordingly, so that both Transformers have input/output dimension $d + n + k$. Therefore, the "last $k$ rows of $Z_{\ell}$" as described in this lemma is equivalent to $Z_3^{out}$ defined in Section 3.1.*

2. *In practice, for both the Laplacian Transformer $\mathsf{TF}^{\mathcal{L}}$ and the Eigenmap Transformer $\mathsf{TF}^{\Psi}$ we found it beneficial to use a multiple parallel heads. One benefit, for instance, is that $H$ heads can simultaneously orthogonalize $H$ columns.*

3. *We loop each layer twice, and the entire Transformer twice in the forward pass, which we find to stabilize training.*

*Proof.* Our proof is primarily based on Lemma 6, Corollary 7, and Lemma 9 of Cheng et al. (2025). For ease of reference, we reproduce the necessary details below.

We will demonstrate that the Transformer is capable of implementing the following block-power-method algorithm (also known as subspace iteration):

$$
\begin{aligned}
\hat{\Phi}_{\ell+1} &= \Phi_{\ell}(\mu I - \Psi^{\top}\Psi) \\
\Phi_{\ell+1} &= QR(\hat{\Phi}_{\ell+1}).
\end{aligned} \tag{17}
$$

The above is initialized at $\Psi_0 \in \mathbb{R}^{k \times n}$ with full column rank. In the above, $QR(\cdot): \mathbb{R}^{k \times n} \to \mathbb{R}^{k \times n}$ returns the $Q$ matrix in the $QR$ decomposition of the input matrix. Under the above, the rows of $\Psi$ converge to the top $k$ eigenvectors of $\mu I - \Psi^{\top}\Psi$, which are equivalent to the bottom $k$ eigenvectors of $\Psi^{\top}\Psi$ by definition of $\mu$.

Concretely, let $Z_{\ell}^{\Psi} \in \mathbb{R}^{n \times n}$ and $Z_{\ell}^{\Phi} \in \mathbb{R}^{k \times n}$ denote the first $n$ rows and last $k$ rows of $Z$ respectively:

$$
Z_{\ell}^{\top} =: \left[ Z_{\ell}^{\Psi^{\top}}, Z_{\ell}^{\Phi^{\top}} \right].
$$

Intuitively, $\Phi_{\ell}$ in the block power method algorithm in (17) is exactly $Z_{\ell}^{\Phi}$.

Our proof proceeds in two steps. First, we verify that there exists a choice of parameters under which the Transformer orthogonalizes the $i^{th}$ column of $Z_{\ell}^{\Phi}$ with respect to all columns $j > i$. The construction is identical to Lemma 8 of Cheng et al. (2025).

Let us choose $s_{\ell} = S_{\ell} = a_{\ell} = b_{\ell} = c_{\ell} = 0$. Let $[A_{\ell}]_{ii} = -1$ and $A_{jj} = 0$ for $j \neq i$. Let $[B_{\ell}]_{jj} = 1$ for $j < i$ and $[B_{\ell}]_{jj} = 0$ for $j \geq i$. Let $C_{\ell} = B_{\ell}$, so that $B_{\ell}^{\top}C_{\ell} = B_{\ell}$. Then we verify that

$$
\begin{aligned}
Z_{\ell+1} &= Z_{\ell} + W_{\ell}^V Z_{\ell} Z_{\ell}^{\top} W_{\ell}^{K^{\top}} W_{\ell}^Q Z_{\ell} \\
&= \begin{bmatrix} Z_{\ell}^{\Psi} \\ Z_{\ell}^{\Phi} \end{bmatrix} - \begin{bmatrix} 0_{n \times n} \\ A_{\ell} Z_{\ell}^{\Phi} Z_{\ell}^{\Phi^{\top}} B_{\ell} Z_{\ell}^{\Phi} \end{bmatrix}.
\end{aligned} \tag{18}
$$

We verify that the $i^{th}$ row of the last term is

$$
\left[ A_{\ell} Z_{\ell}^{\Phi} Z_{\ell}^{\Phi^{\top}} B_{\ell} Z_{\ell}^{\Phi} \right]_i = \sum_{j=1}^{i-1} \left\langle Z_{\ell\ i}^{\Phi}, Z_{\ell\ j}^{\Phi} \right\rangle Z_{\ell\ j}^{\Phi}.
$$

Since each row of $Z_{\ell}^{\Phi}$ has unit norm due to normalization in (15), the above expression is exactly the projection of the $i^{th}$ row onto the subspace spanned by rows $i + 1, \ldots, k$. Consequently, (18) orthogonalizes the $i^{th}$ row of $Z_{\ell}^{\Phi}$ wrt rows $i + 1, \ldots, k$. Repeating the same construction for each row $i$, over $k$ layers, we exactly implement the QR decomposition of $Z_{\ell}^{\Phi}$.

Finally, we verify that there exists a construction under which the Transformer implements the first step of (17). This step is simple: first, in all constructions in this proof, we will pick $a_{\ell} = s_{\ell} = 0$.

Consequently, $Z_\ell^\Psi$ is never updated, and $Z_\ell^\Psi = \Psi$ for all $\ell$. Let us now choose $a_\ell = s_\ell = 0$, $b_\ell = c_\ell = 1$ and $A_\ell = I_{k \times k}$, $B_\ell = C_\ell = 0_{k \times k}$, and $S_\ell = (\mu - 1)I_{k \times k}$. We verify that

$$
\begin{aligned}
Z_{\ell+1} &= Z_\ell + W_\ell^V Z_\ell Z_\ell^\top W_\ell^{K^\top} W_\ell^Q Z_\ell \\
&= \begin{bmatrix} \Psi \\ Z_\ell^\Phi \end{bmatrix} - \begin{bmatrix} 0_{n \times n} \\ Z_\ell^\Phi \Psi^\top \Psi \end{bmatrix} + \begin{bmatrix} 0_{n \times n} \\ (\mu - 1)Z_\ell^\Phi \end{bmatrix} \\
&= \begin{bmatrix} \Psi \\ Z_\ell^\Phi (\mu I - \Psi^\top \Psi) \end{bmatrix}.
\end{aligned}
$$

The above is exactly the first step of (16). We have thus verified Transformer constructions for implementing both steps of (16). $\qquad\square$

## B  DERIVATION OF THE GD UPDATE EQUATION FOR ICL WITH CATEGORICAL OBSERVATIONS

Our derivation is based on the assumption that $f(\phi)$ resides in a reproducing kernel Hilbert space (RKHS) Schölkopf & Smola (2002), but the setup extends to softmax-based attention kernels as well Wang et al. (2025). From the RKHS perspective, let $f(\phi) = A\psi(\phi)$, with $\psi(\phi)$ a *fixed* mapping of features $\phi$ to a Hilbert space, and the parameters acting in that space are associated with matrix $A$.

Assuming we have $m$ labeled samples, indexed $i = 1, \ldots, m$, the cross-entropy cost function for inferring the parameters $A \in \mathbb{R}^{d' \times K}$ (let $K$ represent the output dimension of $\psi(\phi)$) may be expressed as

$$
\begin{aligned}
\mathcal{L}(A) &= -\frac{1}{m} \sum_{i=1}^m \log \left[ \frac{\exp[w_{y_i}^T(A\psi(\phi^{(i)}))]}{\sum_{c=1}^C \exp[w_c^T(A\psi(\phi^{(i)}))]} \right] \\
&= -\frac{1}{m} \sum_{i=1}^m [w_{y_i}^T A\psi(\phi^{(i)}) - \log \sum_{c=1}^C \exp(w_c^T A\psi(\phi^{(i)}))].
\end{aligned}
\tag{19}
$$

Let $a^{(j)} \in \mathbb{R}^K$ represent the $j$th row of $A$. Taking the gradient of $\mathcal{L}$ wrt $a^{(j)}$:

$$
\begin{aligned}
\nabla_{a^{(j)}} \mathcal{L} &= -\frac{1}{m} \sum_{i=1}^m \left[ w_{y_i}(j)\psi(\phi^{(i)}) - \frac{\sum_{c=1}^C \exp[w_c^T(A\psi(\phi^{(i)}))]w_c(j)\psi(\phi^{(i)})}{\sum_{c'=0}^C \exp(w_{c'}^T(A\psi(\phi^{(i)})))} \right] \\
&= -\frac{1}{m} \sum_{i=1}^m \left[ w_{y_i}(j) - \frac{\sum_{c=1}^C \exp[w_c^T f^{(i)}]w_c(j)}{\sum_{c'=0}^C \exp(w_{c'}^T f^{(i)})} \right] \psi(\phi^{(i)}) \\
&= -\frac{1}{m} \sum_{i=1}^m \left[ w_{y_i}(j) - \mathbb{E}(w(j)|f^{(i)}) \right] \psi(\phi^{(i)}).
\end{aligned}
\tag{20}
$$

The gradient update step for $a^{(j)}$ is

$$
\begin{aligned}
a_{\ell+1}^{(j)} &= a_\ell^{(j)} - \alpha \nabla_{a^{(j)}} \mathcal{L} \\
&= a_\ell^{(j)} + \frac{\alpha}{m} \sum_{i=1}^m \left[ w_{y_i}(j) - \mathbb{E}(w(j)|f^{(i)}) \right] \psi(\phi^{(i)}).
\end{aligned}
$$

Using the GD update rules for $\{a^{(j)}\}_{j=1,d'}$, we have

$$
\begin{aligned}
f_{\ell+1}^{(j)} &= \begin{pmatrix} (a_{\ell+1}^{(1)})^T \psi(\phi^{(j)}) \\ \vdots \\ (a_{\ell+1}^{(d')})^T \psi(\phi^{(j)}) \end{pmatrix} \\
\\
&= f_\ell^{(j)} + \frac{\alpha}{m} \sum_{i=1}^m \left[ w_{y_i} - \mathbb{E}(w|f_\ell^{(i)}) \right] \kappa(\phi^{(i)}, \phi^{(j)}).
\end{aligned}
\tag{21}
$$

## C TRANSFORMER FOR CATEGORICAL ICL

The Transformer design for ICL (referred to as $\mathsf{TF}_{sup}$ in the main paper) is based on the GD update equations developed in Section B. The input to the transformer at layer $\ell$ is

$$e_\ell^{(i)} = \begin{pmatrix} f_\ell^{(i)} \\ \mathbb{E}(w|f_\ell^{(i)}) \\ w_{y_i} \\ \phi^{(i)} \end{pmatrix} \tag{22}$$

Within $e_\ell^{(i)}$, the vector component $f_\ell^{(i)}$ is iteratively updated with increasing layer index $\ell$, with the update manifested by each self-attention layer. The expectation $\mathbb{E}(w|f_\ell^{(i)})$ is updated by each MLP layer. Vector components $f_\ell^{(i)}$ and $\mathbb{E}(w|f_\ell^{(i)})$ occupy what we term as *computational scratch space*. The features $\phi^{(i)}$ and embedding vector $w_{y_i}$ represent the encoding of the data $(\phi^{(i)}, y_i)$, and the portion of $e_\ell^{(i)}$ occupied by $(\phi^{(i)}, w_{y_i})$ remains fixed at all Transformer layers.

Each attention block consists of a self-attention layer, composed of two attention heads; one of these attention heads implements $f_\ell^{(i)} \to f_{\ell+1}^{(i)}$ like above (for which $\mathbb{E}(w|f_\ell^{(i)})$ is needed), and the second attention head erases $\mathbb{E}(w|f_\ell^{(i)})$, preparing for its update by the subsequent MLP layer.

### C.1 SELF-ATTENTION LAYER

In matrix form, the input at layer $\ell$ is

$$\begin{pmatrix} f_\ell^{(1)} & \cdots & f_\ell^{(m)} & f_\ell^{(m+1)} \\ \mathbb{E}(w|f_\ell^{(1)}) & \cdots & \mathbb{E}(w|f_\ell^{(m)}) & \mathbb{E}(w|f_\ell^{(m+1)}) \\ w_{y_1} & \cdots & w_{y_m} & 0_{d'} \\ \phi_1 & \cdots & \phi_m & \phi_{m+1} \end{pmatrix} \tag{23}$$

We assume that there are $m$ labeled samples, index $i = 1, \ldots, m$, and sample $m + 1$ is unlabeled. For notational simplicity we only consider one unlabeled sample, corresponding to sample $m+1$. In the semi-supervised setting, all unlabeled samples are treated the same was as sample $m + 1$ shown here.

The update equation for $f_{\ell+1}^{(i)}$ is given by

$$f_{\ell+1}^{(i)} = f_\ell^{(i)} + \Delta f_\ell^{(i)} \tag{24}$$

where

$$\Delta f_\ell^{(i)} = \frac{\alpha}{m} \sum_{j=1}^{m} (w_{y_i} - \mathbb{E}(w|f_\ell^{(i)})) \kappa(\phi^{(i)}, \phi^{(j)}) \tag{25}$$

#### C.1.1 SELF-ATTENTION HEAD 1

We design $W_K^{(1)}$, $W_Q^{(1)}$, and $W_V^{(1)}$ such that

$$W_K^{(1)} e_\ell^{(i)} = (0_{d'}, 0_{d'}, 0_{d'}, \phi^{(i)})^T \tag{26}$$

$$W_Q^{(1)} e_\ell^{(j)} = (0_{d'}, 0_{d'}, 0_{d'}, \phi^{(j)})^T \tag{27}$$

$$W_V^{(1)} e_\ell^{(i)} = (\frac{\alpha}{m}[w_{y_i} - \mathbb{E}(w|f_\ell^{(i)})], 0_d, 0_{d'}, 0_{d'})^T \tag{28}$$

The output of this first attention head, at position $j \in \{1, \ldots, m + 1\}$ is

$$(\frac{\alpha}{m} \sum_{j=1}^{m} (w_{y_j} - \mathbb{E}(w|f_\ell^{(j)})) \kappa(\phi^{(i)}, \phi^{(j)}), 0_d, 0_{d'}, 0_{d'})^T \tag{29}$$

The output of this first attention head at this first attention layer (before adding the skip connection) is

$$O^{(1)} = \begin{pmatrix} \Delta f_\ell^{(1)} & \dots & \Delta f_\ell^{(m)} & \Delta f_\ell^{(m+1)} \\ 0_d & \dots & 0_d & 0_d \\ 0_{d'} & \dots & 0_{d'} & 0_{d'} \\ 0_{d'} & \dots & 0_{d'} & 0_{d'} \end{pmatrix} \tag{30}$$

### C.1.2 SELF-ATTENTION HEAD 2

With the second attention head we want to add $(0_d, -\mathbb{E}(w|f_\ell^{(j)}), 0_{d'}, 0_{d'})^T$ from position $j$, so we clear out the prior expectation. This will provide "scratch space" into which, with the next attention layer type, we will update the expectation, using $f_{\ell+1}^{(j)}$. To do this, we design $W_Q^{(2)}$ and $W_K^{(2)}$ such that

$$W_K^{(2)} e_\ell^{(i)} = \lambda(0_{d'}, 0_{d'}, 0_{d'}, \phi^{(i)})^T \tag{31}$$

$$W_Q^{(2)} e_\ell^{(j)} = \lambda(0_{d'}, 0_{d'}, 0_{d'}, \phi^{(j)})^T \tag{32}$$

where $\lambda >> 1$. With an RBF kernel, for example (similar things will happen with softmax), if $\lambda$ is very large,

$$\kappa(W_K^{(2)} e_\ell^{(i)}, W_Q^{(2)} e_\ell^{(j)}) = \delta_{i,j} \tag{33}$$

where $\delta_{i,j} = 1$ if $i = j$, and it's zero otherwise.

The value matrix is designed as

$$W_V^{(2)} e_\ell^{(i)} = (0_d, \mathbb{E}(w|f_\ell^{(i)}), 0_{d'}, 0_{d'})^T \tag{34}$$

The output of this head is

$$O^{(2)} = \begin{pmatrix} 0_d & \dots & 0_d & 0_d \\ \mathbb{E}(w|f_\ell^{(1)}) & \dots & \mathbb{E}(w|f_\ell^{(m)}) & \mathbb{E}(w|f_\ell^{(m+1)}) \\ 0_{d'} & \dots & 0_{d'} & 0_{d'} \\ 0_{d'} & \dots & 0_{d'} & 0_{d'} \end{pmatrix} \tag{35}$$

We then add $P^{(1)}O^{(1)} + P^{(2)}O^{(2)}$, with $P^{(1)}$ and $P^{(2)}$ designed so as to yield the cumulative output of the attention

$$O^{(\text{total})} = \begin{pmatrix} \Delta f_\ell^{(1)} & \dots & \Delta f_\ell^{(m)} & \Delta f_\ell^{(m+1)} \\ -\mathbb{E}(w|f_\ell^{(1)}) & \dots & -\mathbb{E}(w|f_\ell^{(m)}) & -\mathbb{E}(w|f_\ell^{(m+1)}) \\ 0_d & \dots & 0_d & 0_d \\ 0_{d'} & \dots & 0_{d'} & 0_{d'} \end{pmatrix} \tag{36}$$

This is now added to the skip connection, yielding the total output of this attention layer as

$$T = \begin{pmatrix} f_{\ell+1}^{(1)} & \dots & f_{\ell+1}^{(m)} & f_{\ell+1}^{(m+1)} \\ 0_{d'} & \dots & 0_{d'} & 0_{d'} \\ w_{y_1} & \dots & w_{y_N} & 0_{d'} \\ \phi_1 & \dots & \phi_m & \phi_{m+1} \end{pmatrix} \tag{37}$$

With the first attention layer, with two heads, we update the functions, and we also erase the prior expectations. In the next attention layer, we update the expectations, and place them in the locations of the prior expectations.

## C.2 Multi-Layer Perceptron (MLP) Layer

The vectors connected to $T$ above will go into the next layer, which will be characterized by a MLP. Ideally, the MLP should implement the function

$$\mathbb{E}(w|f_{\ell+1}^{(i)}) = \sum_{c=1}^{C} w_c \left[ \frac{\exp(w_c^T f_{\ell+1}^{(i)})}{\sum\limits_{c'=1}^{C} \exp(w_{c'}^T f_{\ell+1}^{(i)})} \right] \tag{38}$$

to be consistent with functional GD. Let $g_\gamma(f_{\ell+1}^{(i)})$ represent an MLP with parameters $\gamma$. The same MLP acts on each of the vectors at positions $i = 1, \ldots, m$, corresponding to the first $m$ columns of $T$, from left. The components of that vector corresponding to $f_{\ell+1}^{(i)}$ are input to $g_\gamma(\cdot)$, and the output is a $d'$-dimensional vector. The output is placed in the position of the zeros in $T$.

At each layer of the Transformer, the form of the function in (38) is the same. Consequently, within the Transformer implementation, we tie the MLP parameters across all Transformer layer. The MLP parameters $\gamma$ are learned via the cross-entropy loss connected to the query $(\phi_{m+1}, y_{m+1})$. While the MLP is motivated via functional GD to implement (38), this is not explicitly imposed when learning all Transformer parameters.

## D Experimental-Setup Details

**Parameter descriptions.**

- **Number of sample points** ($n = 100$): The number of data points sampled from each manifold. These points form the vertices of our graph construction.
- $k$-**nearest neighbours** ($k = 6$): The number of nearest neighbors used when constructing the adjacency graph for each point. This parameter directly influences the connectivity of the graph and the subsequent Laplacian matrix construction.
- **RBF kernel width** ($\alpha = 10$): The scaling parameter for the radial basis function kernel used to compute weights in the adjacency matrix. In the code, this appears as:

  ```
  A_rbf = torch.exp(-scale_rbf*dist_sq)  # scale_rbf = 10
  ```

- **Eigenvectors exported** ($k_{\text{feat}} = 4$): The number of eigenvectors of the Laplacian matrix that are extracted and used as features. Although the Laplacian has $n$ eigenvectors, we only use the first 4 corresponding to the smallest eigenvalues, as these capture the most important structural information of the manifold.
- **Manifold transformations**: The isotropic scale factor, planar rotation angle, and planar translation are randomly applied to the embedded coordinates after sampling. These transforms can be found in the implementation code, for example:

  ```
  # Apply random scaling
  scale_factor = np.random.uniform(0.02, 0.1)
  a_translated = a * scale_factor

  # Apply random rotation
  theta = np.random.uniform(0, 2 * np.pi)
  rotation_matrix = torch.tensor([
      [np.cos(theta), -np.sin(theta), 0],
      [np.sin(theta), np.cos(theta), 0],
      [0, 0, 1]
  ], dtype=torch.float32)
  a_rotated = torch.matmul(a_translated, rotation_matrix)

  # Apply random translation
  translation_x = np.random.uniform(-1, 1)
  translation_y = np.random.uniform(-1, 1)
  translation = torch.tensor([[[translation_x, translation_y, 0]]])
  a_translated = a_rotated + translation
  ```

The random scale, rotation, and translation are applied *after* sampling and therefore leave all intrinsic geometric properties unchanged. The scale factor uniformly multiplies all coordinates, preserving relative distances and angles. These transformations ensure that our model learns the inherent structure of the manifold rather than relying on specific coordinate representations.

# E SYNTHETIC MANIFOLDS

## E.1 CLASSIC MANIFOLDS IN $\mathbb{R}^3$

Formally, let $(\mathcal{M}, g)$ be any of the five smooth, connected Riemannian manifolds illustrated in Figure 2. Each manifold admits (i) an elementary chart $\Phi : U \subset \mathbb{R}^m \to \mathbb{R}^3$ that we list below, and (ii) a closed-form intrinsic distance $d_\mathcal{M}$. For every experiment we draw $n = 100$ points $\{\tilde{x}_i\}_{i=1}^n \subset \mathbb{R}^3$ by sampling the chart parameters uniformly and then applying an independent random rigid motion (isotropic scale, 3-D rotation, planar translation); this removes global cues while preserving the intrinsic geometry that our model must discover. A centre index $\hat{i} \sim \text{Unif}\{1, \dots, n\}$ is selected and binary labels are assigned via

$$y_i = \begin{cases} 1, & d_\mathcal{M}(\tilde{x}_i, \tilde{x}_{\hat{i}}) < \tau_\mathcal{M}, \\ 0, & \text{otherwise}, \end{cases}$$

so that the positive class is exactly the geodesic ball of radius $\tau_\mathcal{M}$ about $\tilde{x}_{\hat{i}}$. The explicit charts, geodesic formulas, and manifold-specific thresholds $\tau_\mathcal{M}$ are collected in the remainder of this section.

### E.1.1 SPHERICAL FAMILY $\mathbf{S}^2$

- Parametrization:

$$\Phi(\theta, \phi) = \big(\sin\theta\cos\phi, \ \sin\theta\sin\phi, \ \cos\theta\big), \qquad (\theta, \phi) \in (0, \pi) \times (0, 2\pi).$$

- Line element:
  $ds^2 = d\theta^2 + \sin^2\theta \, d\phi^2$.

- Geodesic distance:
  Great-circle length $d_{S^2}(p, q) = \arccos\langle p, q \rangle$.

- Labeling threshold:
  $\tau_\mathrm{S} = \pi/3$ radians. This corresponds to approximately 60 degrees of separation on the sphere's surface, creating a spherical cap that covers roughly 25% of the sphere's surface area.

### E.1.2 CYLINDER FAMILY

- Parametrization:

$$\Phi(\theta, z) = (r\cos\theta, \ r\sin\theta, \ z), \qquad r \equiv 1.$$

- Line element:
  $ds^2 = r^2 d\theta^2 + dz^2$.

- Distance:
  $d_{\mathrm{cyl}}^2 = (r\,\Delta\theta)^2 + (z_1 - z_2)^2$.

- Labeling threshold:
  $\tau_{\mathrm{cyl}} = 1.0$. Given the unit radius, this threshold allows the geodesic distance to extend approximately 1 radian around the cylinder circumference and 1 unit along its height.

### E.1.3 CONE FAMILY

- Parametrization:

$$\Phi(s, \theta) = \big(s\sin\alpha\cos\theta, \ s\sin\alpha\sin\theta, \ s\cos\alpha\big), \quad \alpha \in (0, \tfrac{\pi}{2}).$$

- Line element:

$$ds^2 = ds^2 + s^2 \sin^2 \alpha \, d\theta^2.$$

- Distance:
  Unfolding to a sector gives

$$d^2_{\text{cone}} = s_1^2 + s_2^2 - 2s_1 s_2 \cos(\sin \alpha \, \Delta\theta).$$

- Labeling threshold:
  $\tau_{\text{cone}} = 0.5$. This creates a circular region on the cone's surface, with the size of the region depending on the cone's apex angle $\alpha$.

### E.1.4 ARCHIMEDEAN SPIRAL (SWISS ROLL)

- Parametrization:
  $\Phi(t) = (t^2 \cos 4\pi t, \ t^2 \sin 4\pi t, \ 1), \ t \in [0, 1]$.
- Line element:
  Speed $\|\dot{\Phi}(t)\| = 2t\sqrt{1 + 4\pi^2 t^2}$; hence

$$ds = 2t\sqrt{1 + 4\pi^2 t^2} \, dt.$$

- Distance:
  Integrating and taking absolute differences yields

$$d_{\text{SR}}(t_1, t_2) = \frac{\left|(1 + 4\pi^2 t_2^2)^{3/2} - (1 + 4\pi^2 t_1^2)^{3/2}\right|}{6\pi^2}.$$

- Labeling threshold:
  The class boundary is defined at $\text{median}(t)$, where points with parameter $t$ less than the median are assigned to class 1, and others to class 0. This creates a natural division of the Swiss roll into inner and outer regions.

### E.1.5 FLAT TORUS $\mathbb{T}^2$

- Parametrization:
  $\Phi(\theta, \phi) = (\theta, \phi, 0), \ (\theta, \phi) \in (0, 2\pi)^2$.
- Line element:
  $ds^2 = d\theta^2 + d\phi^2$.
- Distance (wrapped Euclidean):

$$d^2_{\mathbb{T}^2} = \Delta\theta^2 + \Delta\phi^2, \qquad \Delta\theta = \min(|\theta_1 - \theta_2|, \ 2\pi - | \cdot |).$$

- Labeling threshold:
  $\tau_{\mathbb{T}^2} = 0.5$. Given the parameter range $(0, 2\pi)^2$, this threshold creates a small circular region on the flat torus, corresponding to approximately 1/8 of the torus surface area.

### E.2 PRODUCT MANIFOLDS

The five classical manifolds introduced in Section E.1 are all low-dimensional and possess relatively simple label boundaries. In practical applications, however, data often reside on much *higher-dimensional* nonlinear spaces whose intrinsic geometry cannot be recovered from any single elementary chart. By taking the Cartesian product of elementary manifolds, we can construct highly complex manifolds in arbitrary dimensions. These product manifolds play an important role in practical applications ranging from computational chemistry (Zhang et al., 2021) and image and 3D structure analysis (Fumero et al., 2021). Evaluating on these products therefore probes the *algorithmic* generalisability of our Transformer—namely, whether it can still recover the manifold structure and correct labels when confronted with far higher ambient dimension and intricate decision boundaries.

**Definition.** For any integer $K \geq 2$ and any ordered tuple $(\mathcal{M}_1, \ldots, \mathcal{M}_K)$ chosen from our base set $\{\text{Sphere, Cylinder, Cone, Spiral}, \mathbb{T}^2\}$ we consider the Cartesian product $\mathcal{P} = \mathcal{M}_1 \times \cdots \times \mathcal{M}_K$. Each factor inherits its Riemannian metric $g_{\mathcal{M}_k}$, and $\mathcal{P}$ carries the standard *product metric* $g = \sum_{k=1}^{K} \pi_k^* g_{\mathcal{M}_k}$, leading to the distance

$$d_\times(P, Q) = \Big(\sum_{k=1}^{K} d_{\mathcal{M}_k}(p_k, q_k)^2\Big)^{1/2}, \qquad P = (p_1, \ldots, p_K), \ Q = (q_1, \ldots, q_K). \quad (39)$$

### E.2.1 STATEMENT AND PROOF OF THEOREM 1

**Theorem 1** (Geodesic distance on a product manifold). *Let $(M_i, g_i)$ be complete Riemannian manifolds with intrinsic distances $d_{M_i}$. Endow $M = M_1 \times \cdots \times M_K$ with the product metric described above. Then for every $p, q \in M$, $d_M(p, q) = \sqrt{\sum_{i=1}^{K} d_{M_i}(\pi_i p, \pi_i q)^2}$.*

Although the formula (39) is intuitive, we were unable to locate an explicit proof in the literature. For completeness, we provide the following proof.

Restatement:

Let $p = (p_1, \ldots, p_k)$ and $q = (q_1, \ldots, q_k)$ be points of $M$. Then

$$d_M(p, q) = \sqrt{d_{M_1}(p_1, q_1)^2 + \ldots + d_{M_k}(p_k, q_k)^2}.$$

*Proof.* Set $d_i := d_{M_i}(p_i, q_i)$ for $1 \leq i \leq k$. We prove the two opposite inequalities.

**Lower bound.** Let $c : [0, 1] \to M$ be any piecewise $C^1$ curve with $c(0) = p$ and $c(1) = q$. Write $c(t) = (c_1(t), \ldots, c_k(t))$ and denote

$$a_i(t) := \|\dot{c}_i(t)\|_{g_i} \quad (\geq 0).$$

Because $g(\dot{c}, \dot{c}) = \sum_{i=1}^{k} a_i(t)^2$,

$$L(c) = \int_0^1 \sqrt{\sum_{i=1}^{k} a_i(t)^2} \, dt.$$

Using the triangle inequality in $\mathbb{R}^k$,

$$\Big\| \int_0^1 (a_1(t), \ldots, a_k(t)) \, dt \Big\|_{\mathbb{R}^k} \leq L(c).$$

But $\int_0^1 a_i(t) \, dt = L(c_i) \geq d_i$ for each $i$, so

$$L(c) \geq \sqrt{\sum_{i=1}^{k} d_i^2}.$$

Taking the infimum over all $c$ yields

$$d_M(p, q) \geq \sqrt{\sum_{i=1}^{k} d_i^2}.$$

**Upper bound.** Fix $\varepsilon > 0$. For each $i$ choose a piecewise $C^1$ curve $\gamma_i : [0, 1] \to M_i$ joining $p_i$ to $q_i$ with constant speed and

$$L(\gamma_i) < d_i + \varepsilon.$$

Define $c : [0, 1] \to M$ by $c(t) := (\gamma_1(t), \ldots, \gamma_k(t))$. Since each $\gamma_i$ has constant speed $a_i := L(\gamma_i)$,

$$g(\dot{c}(t), \dot{c}(t)) = \sum_{i=1}^{k} a_i^2 \quad \text{(independent of } t\text{)},$$

so

$$L(c) = \sqrt{\sum_{i=1}^{k} a_i^2} \; < \; \sqrt{\sum_{i=1}^{k} (d_i + \varepsilon)^2}.$$

Letting $\varepsilon \downarrow 0$ gives the reverse inequality

$$d_M(p, q) \; \leq \; \sqrt{\sum_{i=1}^{k} d_i^2}.$$

$\square$

**Remark 4.** *If every $(M_i, g_i)$ is complete, the Hopf–Rinow theorem guarantees the existence of minimizing geodesics; in that case the curve constructed in the upper bound can be chosen geodesic in every factor, giving an explicit length minimizer in the product manifold.*

### E.3    IMAGE MANIFOLD

To evaluate our approach on high-dimensional, we created image manifolds using Stable Diffusion v1.5 Rombach et al. (2022). These manifolds represent a significant increase in complexity and dimensionality compared to our synthetic manifolds, testing the transfer capabilities of our semi-supervised in-context learning method.

#### E.3.1    IMAGE MANIFOLD GENERATION

Our image manifolds are constructed using spherical linear interpolation (slerp) in the latent space of Stable Diffusion v1.5. For each manifold:

1. We sample two independent random latent vectors from a standard normal distribution.
2. We perform spherical linear interpolation between these two vectors, generating 100 evenly spaced intermediate points.
3. Each interpolated latent vector is decoded into a 500×500 pixel RGB image using the Stable Diffusion model without classifier guidance.
4. This process is repeated for 950 different random seed pairs, creating 950 diverse image manifolds.

As in our other experiments, there are $n = 100$ total samples connected with a given manifold, here corresponding to 100 images. The smooth transitions between images in each manifold confirm that the latent space of the diffusion model indeed has a manifold structure. This allows us to test our methods on natural image data while maintaining control over the underlying manifold geometry.

#### E.3.2    DATA PROCESSING

The image data processing involves several specific steps:

- **Downsampling**: All 500×500 images are downsampled to 32×32 pixels. This reduces the dimensionality from 250,000 to 1,024 dimensions per image while preserving essential visual features.
- **Grayscale conversion**: All images are converted to grayscale to simplify processing.
- **Normalization**: Pixel values are first normalized to the range [0,1] by dividing by 255.
- **Pixel scaling**: A pixel scaling factor of 0.001 is applied, reducing the normalized pixel values to the range [0, 0.001]. This scaling is specifically designed to match the distance scale of image manifolds with our synthetic manifolds, enabling effective transfer of models trained on synthetic data to image data. Without this scaling factor, the distances between image points would have a significantly different distribution compared to synthetic manifolds, hindering generalization.

The relevant code for image processing is:

```
# Downsample from 500×500 to 32×32
img = img.resize((32, 32), Image.LANCZOS)

# Convert to tensor and normalize to [0, 1]
img_tensor = torch.from_numpy(np.array(img)).float() / 255.0

# Apply pixel scale factor of 0.001
img_tensor = img_tensor / 1000.0
```

### E.3.3 TRAIN-TEST SPLIT

We divide the 950 image manifolds between training and testing:

- **Training manifolds**: Prompts with IDs 1-500 (first 500 manifolds)
- **Test manifolds**: Prompts with IDs 501+ (remaining manifolds)

This split ensures that models are evaluated on previously unseen interpolation sequences, testing true generalization rather than memorization.

### E.3.4 GRAPH CONSTRUCTION

For each image manifold:

- We compute pairwise Euclidean distances between flattened image vectors.
- These distances can be (optionally) scaled using the parameter `--image_scale 0.001`.
- An RBF kernel is applied to convert distances to similarities: $A_{ij} = \exp(-\alpha \cdot d_{ij}^2)$. $\alpha$ has the same scale 10 here as for our other manifolds.
- A 4-nearest neighbors graph is constructed for each manifold.
- The normalized Laplacian is computed from this graph as for our other manifolds.

### E.3.5 LABELING

For image manifolds, we use a position-based labeling scheme:

- The first 50 images in each interpolation sequence are assigned to class 0.
- The last 50 images are assigned to class 1.

This creates a natural separation based on position in the interpolation sequence, dividing each manifold into two equal parts. Only $m \ll n$ of the images are labeled as given to our Transformer.

### E.3.6 VISUAL REPRESENTATION

To visualize the manifold structure of our image datasets, we present two examples of complete 10×10 image grids, each showing all $n = 100$ interpolated frames from different manifolds. These grids demonstrate the smooth transitions created by the spherical linear interpolation in Stable Diffusion's latent space.

These examples illustrate several important aspects of our image manifolds:

1. **Continuous transitions:** Each row shows gradual changes without abrupt shifts, supporting the assumed smooth manifold structure.

2. **Semantic coherence:** The interpolations maintain thematic consistency while traversing the latent space, with changes in perspective, detail, composition, and style.

3. **Natural class division:** The first 50 images (top 5 rows) and last 50 images (bottom 5 rows) form our two classes, with the most significant changes often occurring near the midpoint of the interpolation.

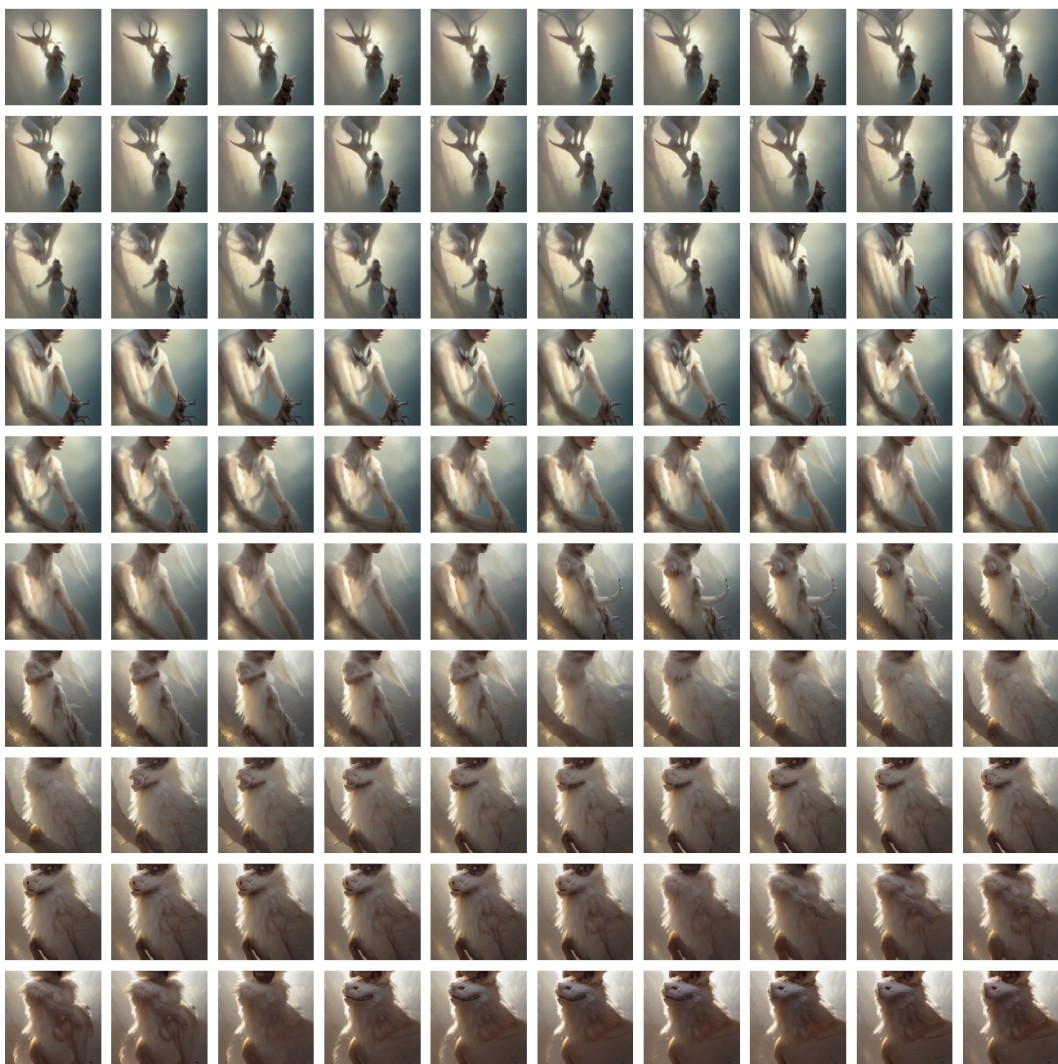

Figure 6: Example image manifold showing interpolation of a deer in atmospheric lighting. The transformation progresses from a distant view of antlers (class 0, top 5 rows) to close-up facial details (class 1, bottom 5 rows), demonstrating how the manifold captures continuous changes in perspective.

4. **Diversity across manifolds:** The three examples demonstrate how our methodology creates varied manifolds across different themes and visual styles, providing a challenging and diverse test set.

Our semi-supervised in-context learning method must navigate these complex visual manifolds, identifying the underlying geometric structure from just a few labeled examples. The ability to transfer knowledge from synthetic 3D manifolds to these high-dimensional image manifolds demonstrates the robust generalization capabilities of our approach.

## F  ADDITIONAL EXPERIMENTAL RESULTS

### F.1  DEPENDENCE ON ICL TRANSFORMER DEPTH

We investigate how the depth of the in-context learning (ICL) Transformer for categorical outcomes (see Section C) affects performance by comparing 1-layer and 2-layer ICL Transformer architectures

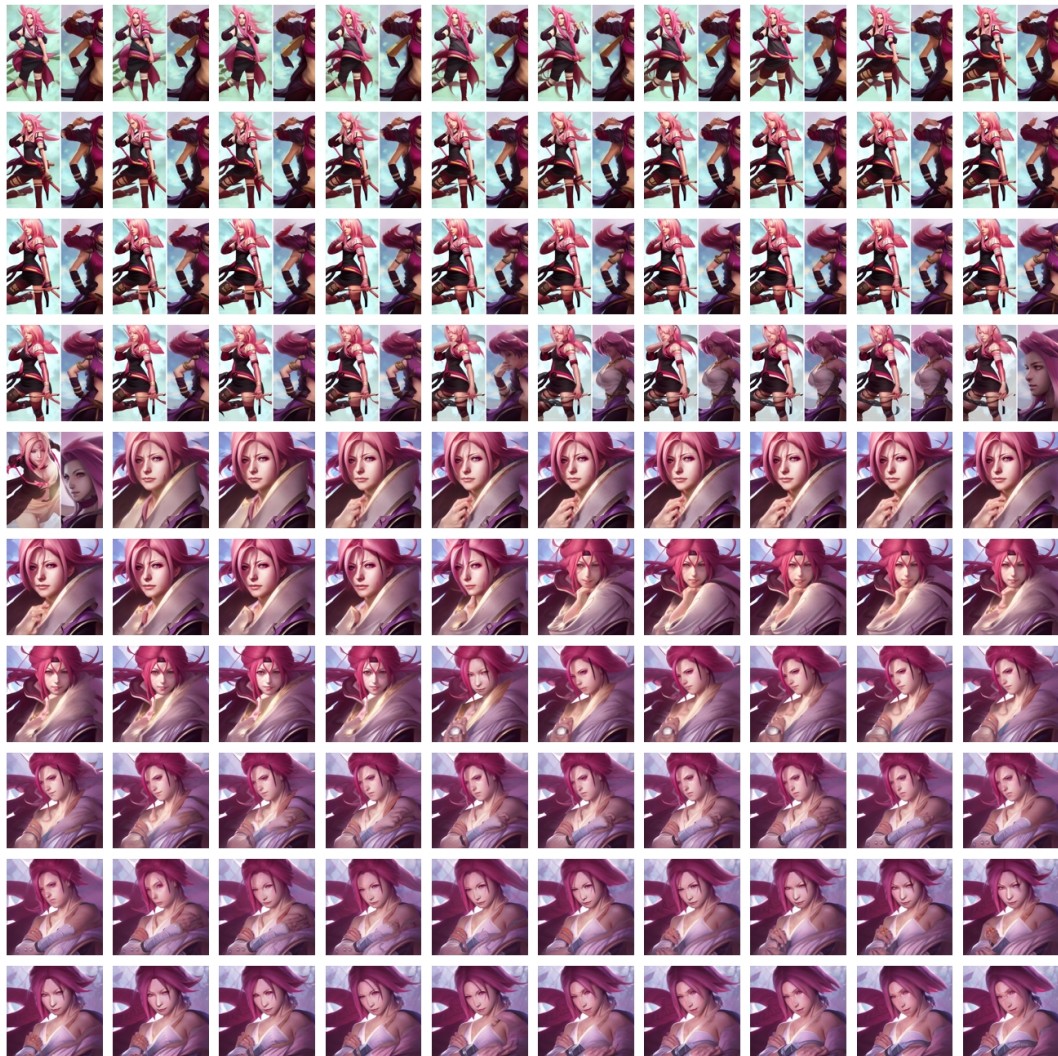

Figure 7: Example image manifold showing interpolation between anime-style character concepts. The progression moves from an action pose with bright colors (class 0, top 5 rows) to increasingly detailed portrait shots (class 1, bottom 5 rows), highlighting the manifold's ability to represent both style and compositional changes.

(in general, we observed little change in performance for deeper models). Figure 8 shows the results of this comparison on the product manifold.

The results demonstrate that the 2-layer ICL architecture consistently outperforms the 1-layer architecture across all labeled data percentages. This supports the theoretical framework in Wang et al. (2025), showing that increased depth allows the model to learn more sophisticated manifold representations. The performance improvement is more pronounced for the RBF kernel, which further emphasizes the kernel's ability to capture non-linear relationships in the data manifold. We observe that for both kernel types, the 2-layer ICL architecture provides a consistent advantage, with RBF kernels outperforming linear counterparts at both depths. This advantage is most notable in the low-label regime (5-15% labeled data), highlighting the importance of architectural capacity when labeled data is scarce.

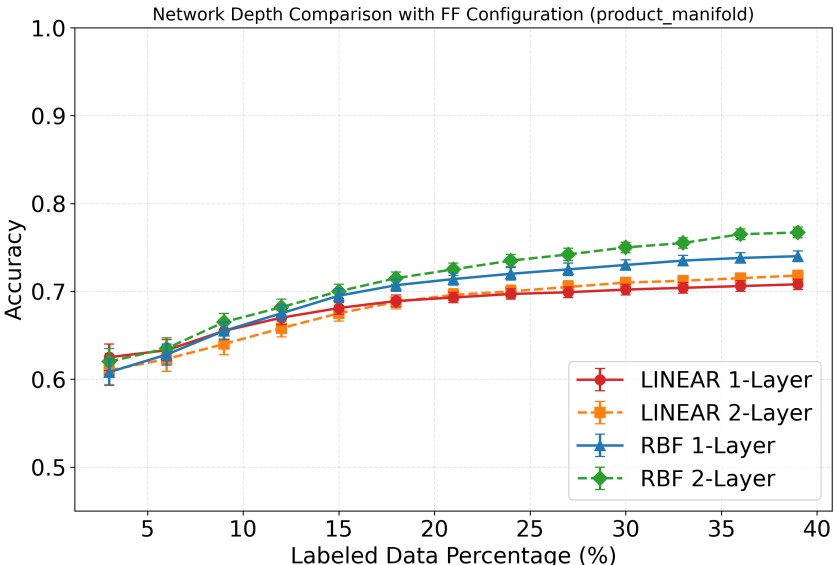

Figure 8: Comparison of 1-layer vs. 2-layer ICL Transformer architectures on the product manifold. The 2-layer architecture shows improved performance, particularly in the low-label regime, suggesting that deeper networks can learn more sophisticated manifold representations.

## F.2    COMPARISON OF KERNEL TYPES ACROSS MANIFOLDS

While the paper focused primarily on results with the RBF kernel on selected manifolds due to space constraints, here we present a comprehensive analysis comparing linear and RBF kernels across all individual manifolds as well as product manifolds.

The results in Figure 9 demonstrate that our approach generalizes well across different manifold types and kernel choices. For each manifold, we observe similar patterns where our end-to-end Transformer consistently outperforms the baselines. While the RBF kernel typically yields better results for most manifolds, the relative performance varies depending on the geometric complexity of the manifold. These results further demonstrate that the semi-supervised Transformer framework we have developed is robust across a variety of manifold structures.

## F.3    IN-DISTRIBUTION VS. OUT-OF-DISTRIBUTION PERFORMANCE

We examine the generalization capabilities of our model by evaluating its performance in both in-distribution (ID) and out-of-distribution (OOD) settings. For ID, the manifold types seen at test were observed when the Transformer was trained, while for OOD the Transformer did not see the class of test manifolds when being trained. For OOD evaluation, we train the model on four manifold types and test on the held-out fifth manifold. Tables 2, 3, and 4 present detailed results for different context sizes.

Our results reveal interesting patterns in generalization capabilities. For simpler manifolds like cylinder, sphere, and torus, we observe that OOD performance is competitive with or sometimes even exceeds ID performance by a large margin. This suggests that training on a diverse set of manifolds provides rich geometric knowledge that transfers well to these simpler structures. The simpler manifolds share similar geometric features, making knowledge transfer more effective.

In contrast, for more complex manifolds like Swiss Roll and Cone, OOD performance is typically lower than ID performance. This is particularly pronounced for the Swiss Roll manifold with the RBF kernel, where OOD performance is significantly lower than ID. This performance gap likely stems from the unique geometric properties of the Swiss Roll that are not well captured by the kernel bandwidth parameters learned from other manifolds. The RBF kernel's bandwidth is particularly sensitive to the intrinsic curvature and density variations present in the Swiss Roll but absent in simpler manifolds.

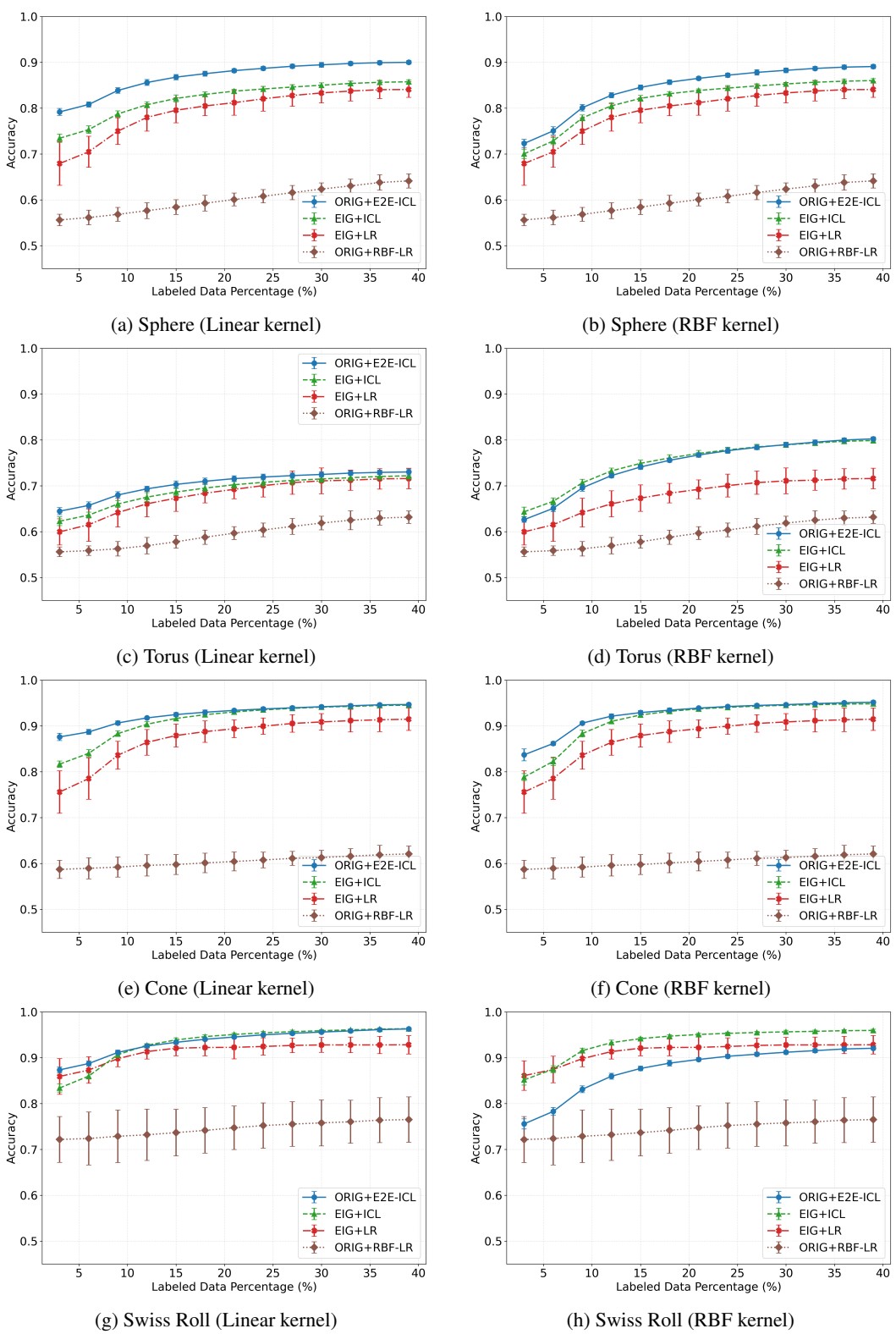

(a) Sphere (Linear kernel)

(b) Sphere (RBF kernel)

(c) Torus (Linear kernel)

(d) Torus (RBF kernel)

(e) Cone (Linear kernel)

(f) Cone (RBF kernel)

(g) Swiss Roll (Linear kernel)

(h) Swiss Roll (RBF kernel)

Figure 9: Comparison of model performance across all individual manifolds with linear and RBF kernels. Our end-to-end Transformer (ORIG+E2E-ICL) consistently outperforms baseline methods across all manifolds and kernel types, with the advantage being particularly pronounced in the low-label regime.

Table 2: Performance Comparison at Context Size 3

| Kernel-Target | EIG+ICL (ID) | EIG+ICL (OOD) | E2E ICL (ID) | E2E ICL (OOD) |
|---|---|---|---|---|
| Linear-Cone | 0.778 ± 0.007 | 0.799 ± 0.007 | 0.859 ± 0.008 | 0.757 ± 0.018 |
| Linear-Cylinder | 0.734 ± 0.007 | 0.790 ± 0.010 | 0.772 ± 0.011 | 0.923 ± 0.007 |
| Linear-Sphere | 0.708 ± 0.009 | 0.788 ± 0.007 | 0.767 ± 0.008 | 0.919 ± 0.005 |
| Linear-Torus | 0.605 ± 0.010 | 0.857 ± 0.009 | 0.627 ± 0.008 | 0.968 ± 0.003 |
| Linear-Swiss Roll | 0.789 ± 0.009 | 0.788 ± 0.008 | 0.854 ± 0.006 | 0.826 ± 0.015 |
| RBF-Cone | 0.731 ± 0.008 | 0.826 ± 0.013 | 0.790 ± 0.013 | 0.753 ± 0.017 |
| RBF-Cylinder | 0.696 ± 0.009 | 0.918 ± 0.006 | 0.725 ± 0.010 | 0.927 ± 0.005 |
| RBF-Sphere | 0.658 ± 0.010 | 0.912 ± 0.006 | 0.679 ± 0.009 | 0.920 ± 0.006 |
| RBF-Torus | 0.614 ± 0.010 | 0.963 ± 0.004 | 0.590 ± 0.006 | 0.969 ± 0.003 |
| RBF-Swiss Roll | 0.811 ± 0.010 | 0.785 ± 0.009 | 0.805 ± 0.010 | 0.694 ± 0.014 |

Table 3: Performance Comparison at Context Size 21

| Kernel-Target | EIG+ICL (ID) | EIG+ICL (OOD) | E2E ICL (ID) | E2E ICL (OOD) |
|---|---|---|---|---|
| Linear-Cone | 0.931 ± 0.004 | 0.955 ± 0.004 | 0.933 ± 0.003 | 0.798 ± 0.015 |
| Linear-Cylinder | 0.861 ± 0.005 | 0.929 ± 0.004 | 0.871 ± 0.006 | 0.935 ± 0.002 |
| Linear-Sphere | 0.837 ± 0.004 | 0.925 ± 0.003 | 0.882 ± 0.004 | 0.930 ± 0.002 |
| Linear-Torus | 0.703 ± 0.008 | 0.964 ± 0.003 | 0.716 ± 0.006 | 0.970 ± 0.001 |
| Linear-Swiss Roll | 0.951 ± 0.003 | 0.952 ± 0.002 | 0.946 ± 0.003 | 0.919 ± 0.009 |
| RBF-Cone | 0.937 ± 0.004 | 0.877 ± 0.009 | 0.939 ± 0.003 | 0.800 ± 0.015 |
| RBF-Cylinder | 0.866 ± 0.005 | 0.934 ± 0.002 | 0.885 ± 0.005 | 0.936 ± 0.002 |
| RBF-Sphere | 0.839 ± 0.003 | 0.929 ± 0.002 | 0.865 ± 0.004 | 0.929 ± 0.003 |
| RBF-Torus | 0.771 ± 0.007 | 0.970 ± 0.001 | 0.768 ± 0.006 | 0.970 ± 0.001 |
| RBF-Swiss Roll | 0.950 ± 0.002 | 0.860 ± 0.007 | 0.940 ± 0.004 | 0.764 ± 0.010 |

Table 4: Performance Comparison at Context Size 39

| Kernel-Target | EIG+ICL (ID) | EIG+ICL (OOD) | E2E ICL (ID) | E2E ICL (OOD) |
|---|---|---|---|---|
| Linear-Cone | 0.947 ± 0.003 | 0.966 ± 0.002 | 0.947 ± 0.003 | 0.803 ± 0.014 |
| Linear-Cylinder | 0.876 ± 0.004 | 0.935 ± 0.002 | 0.885 ± 0.004 | 0.935 ± 0.002 |
| Linear-Sphere | 0.857 ± 0.004 | 0.930 ± 0.003 | 0.901 ± 0.004 | 0.929 ± 0.002 |
| Linear-Torus | 0.724 ± 0.007 | 0.969 ± 0.002 | 0.730 ± 0.006 | 0.970 ± 0.001 |
| Linear-Swiss Roll | 0.964 ± 0.002 | 0.964 ± 0.003 | 0.964 ± 0.002 | 0.947 ± 0.009 |
| RBF-Cone | 0.949 ± 0.002 | 0.877 ± 0.007 | 0.952 ± 0.003 | 0.804 ± 0.016 |
| RBF-Cylinder | 0.882 ± 0.004 | 0.935 ± 0.002 | 0.902 ± 0.005 | 0.935 ± 0.002 |
| RBF-Sphere | 0.860 ± 0.004 | 0.930 ± 0.002 | 0.892 ± 0.004 | 0.930 ± 0.002 |
| RBF-Torus | 0.800 ± 0.006 | 0.970 ± 0.002 | 0.804 ± 0.004 | 0.970 ± 0.001 |
| RBF-Swiss Roll | 0.960 ± 0.002 | 0.868 ± 0.007 | 0.954 ± 0.003 | 0.775 ± 0.009 |

When training exclusively on one manifold type, the model may specialize excessively to the spectral decomposition of that manifold's Laplacian matrix. This results in spectral features with lower bias but significantly higher variance upon finite-sample re-estimation—characteristic of overfitting. Exposure to multiple manifold types appears to promote more robust spectral feature learning that generalizes better.

## F.4 IMAGE MANIFOLD TRANSFER PERFORMANCE

The ability to transfer knowledge from synthetic manifolds to high-dimensional image manifolds is a crucial test of our approach's robustness. Table 5 presents results for different combinations of training manifolds when evaluating on image manifolds.

Table 5: Performance Comparison for RBF Kernel on Image Manifolds at Different Context Sizes

| Context | Training Manifolds | EIG+ICL (ID) | EIG+ICL (OOD) | E2E ICL (ID) | E2E ICL (OOD) |
|---|---|---|---|---|---|
| 3 | cone,cylinder,sphere,swiss_roll | $0.700 \pm 0.008$ | $0.555 \pm 0.005$ | $0.730 \pm 0.007$ | $0.557 \pm 0.007$ |
| | cone,cylinder,torus,swiss_roll | $0.700 \pm 0.006$ | $0.555 \pm 0.005$ | $0.730 \pm 0.009$ | $0.600 \pm 0.007$ |
| | cone,sphere,torus,swiss_roll | $0.700 \pm 0.009$ | $0.555 \pm 0.005$ | $0.730 \pm 0.006$ | $0.596 \pm 0.006$ |
| | cylinder,sphere,torus,swiss_roll | $0.700 \pm 0.007$ | $0.557 \pm 0.009$ | $0.730 \pm 0.008$ | $0.543 \pm 0.005$ |
| | cone,cylinder,sphere,torus | $0.700 \pm 0.005$ | $0.555 \pm 0.007$ | $0.730 \pm 0.010$ | $0.524 \pm 0.004$ |
| 21 | cone,cylinder,sphere,swiss_roll | $0.850 \pm 0.007$ | $0.657 \pm 0.008$ | $0.880 \pm 0.006$ | $0.686 \pm 0.005$ |
| | cone,cylinder,torus,swiss_roll | $0.850 \pm 0.009$ | $0.657 \pm 0.008$ | $0.880 \pm 0.005$ | $0.744 \pm 0.004$ |
| | cone,sphere,torus,swiss_roll | $0.850 \pm 0.006$ | $0.657 \pm 0.008$ | $0.880 \pm 0.008$ | $0.745 \pm 0.003$ |
| | cylinder,sphere,torus,swiss_roll | $0.850 \pm 0.005$ | $0.652 \pm 0.006$ | $0.880 \pm 0.009$ | $0.650 \pm 0.005$ |
| | cone,cylinder,sphere,torus | $0.850 \pm 0.008$ | $0.612 \pm 0.008$ | $0.880 \pm 0.007$ | $0.584 \pm 0.004$ |
| 39 | cone,cylinder,sphere,swiss_roll | $0.900 \pm 0.006$ | $0.689 \pm 0.005$ | $0.930 \pm 0.008$ | $0.740 \pm 0.006$ |
| | cone,cylinder,torus,swiss_roll | $0.900 \pm 0.005$ | $0.689 \pm 0.005$ | $0.930 \pm 0.007$ | $0.781 \pm 0.004$ |
| | cone,sphere,torus,swiss_roll | $0.900 \pm 0.008$ | $0.689 \pm 0.005$ | $0.930 \pm 0.005$ | $0.786 \pm 0.004$ |
| | cylinder,sphere,torus,swiss_roll | $0.900 \pm 0.009$ | $0.688 \pm 0.006$ | $0.930 \pm 0.006$ | $0.710 \pm 0.008$ |
| | cone,cylinder,sphere,torus | $0.900 \pm 0.007$ | $0.626 \pm 0.006$ | $0.930 \pm 0.009$ | $0.616 \pm 0.005$ |

The results demonstrate that our end-to-end semi-supervised Transformer achieves remarkable zero-shot transfer to image manifolds despite being trained solely on synthetic manifolds. Several important observations emerge:

The combination of training manifolds significantly impacts transfer performance. Including both Swiss Roll and torus in the training set appears to consistently yield the best transfer results, likely because these manifolds capture certain intrinsic properties present in image manifolds better than other combinations.

Secondly, there is a clear relationship between context size and transfer performance. As the context size increases from 3 to 39, the transfer performance improves substantially across all training configurations. This suggests that the model's ability to leverage contextual information generalizes well from synthetic to image manifolds.

Finally, our end-to-end approach (E2E ICL) consistently outperforms the baseline EIG+ICL model in both ID and OOD settings, confirming that the learned representation is more effective than using ground-truth eigenvectors directly in diffcult problems.

## F.5 IMAGENET100 EXPERIMENTS WITH 39% LABELS

We additionally report results on ImageNet100 with a higher label ratio of 39%.

**Remark 5** (Separation Score). *The separation score is defined as the difference between average intra-class similarity and inter-class similarity Caron et al. (2018); Oord et al. (2018):*

$$Sep = \frac{1}{|\mathcal{C}|} \sum_{c \in \mathcal{C}} \mathbb{E}_{x_i, x_j \in c} \left[ sim(x_i, x_j) \right] - \mathbb{E}_{x \in c, \, y \in c', \, c \neq c'} [sim(x, y)],$$

*where $sim(\cdot, \cdot)$ denotes cosine similarity.*

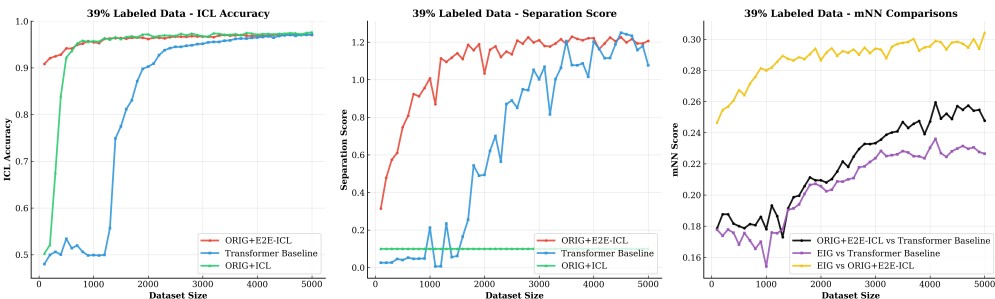

Figure 10: In-context learning on ImageNet100 with $39\%$ labels and dataset size $5000$. Each dataset consists of 200 batches, with each batch containing 100 samples (50 images per class) represented by VGG-29 features. The left column reports ICL accuracy, and the right column reports the separation score (intra-class minus inter-class similarity, following Caron et al. (2018); Oord et al. (2018)). We compare three methods: **Orig+E2E-ICL** (red, PE from our model), a **Transformer baseline** (blue, PE from the final transformer layer Min et al. (2022)), and **Eig+ICL** (green, features pre-extracted from VGG-29).

### F.6 ADDITIONAL EXPERIMENTAL DETAILS

All models were implemented in PyTorch and trained on a Tesla V100 PCIe 16 GB GPU. The complete set of experiments required roughly 2 hours of computation time. All source code is available here: GitHub.

### F.7 MODELS UNDER EVALUATION

We denote models by *[Data+Model]*, where *Data* specifies the form of input features, and *Model* specifies the classifier applied to the unlabeled samples.

**ORIG+E2E-ICL (OURS)** The raw *original* coordinate matrix is fed directly into our **end-to-end Transformer**. All three modules—the Laplacian predictor, eigenvector extractor, and in-context GD head— are trained jointly with a combined loss. (See Appendix A for architectural details.)

**EIG+ICL** The ICL head is supplied with the *ground-truth* bottom-$k$ Laplacian eigenvectors, computed offline. This removes representation learning, so the gap to ORIG+E2E-ICL directly quantifies the benefit of learned representations.

**EIG+LR** A standard $\ell_2$-regularized logistic regression refit on the *ground-truth* eigenvectors at each label ratio.

**ORIG+RBF-LR** Kernel logistic regression on the *original* coordinates using the Gaussian kernel $\kappa_{\mathrm{RBF}}(x, x') = \exp(-\gamma \|x - x'\|_2^2)$ Schölkopf & Smola (2002). The decision function is $\sigma\left(\sum_{j=1}^{n} \alpha_j \kappa_{\mathrm{RBF}}(x_i, x_j)\right)$ with an $\ell_2$ penalty on $\boldsymbol{\alpha}$, providing a nonlinear classifier refit at each label ratio.

**ORIG+ICL** The raw *original* coordinate matrix is fed directly into our in-context GD head, without any additional learned embedding module. This serves as a control to measure the contribution of the learned PE module in ORIG+E2E-ICL.

### F.8 ABLATION ON LAP (LAPLACIAN PREDICTOR) AND PE (EIGENVECTOR EXTRACTOR)

**Experimental Setup:** We replace the RBF activation in the Laplacian-forming module ($\phi_{\mathrm{lap}}$) with a linear mapping (no activation) and additionally compare a **1-layer** variant of our Laplacian and positional embedding (PE) modules. All experiments are conducted on the 5-factor product manifold task (train and test on the same manifold family), averaged over 10 runs per context size.

**Note on "1-layer" configuration:** The "1-layer" Lap+PE variant refers to a non-iterative version of our spectral representation module, which performs only a single forward pass to compute the Laplacian and positional embeddings, without iterative refinement. By contrast, the full in-context

Transformer iteratively refines the Laplacian and PE features through multiple forward passes within each Transformer block. As shown below, the single-pass configuration produces weaker manifold representations, confirming the importance of iterative refinement.

Table 6: Ablation on Laplacian and positional embedding modules using product-manifold data (5 factors).

| Context Size | Lap+PE (1 Layer) | Our Model | Lap (Linear) |
|---|---|---|---|
| 3 | $0.534 \pm 0.008$ | $\mathbf{0.663 \pm 0.031}$ | $0.479 \pm 0.009$ |
| 5 | $0.548 \pm 0.016$ | $\mathbf{0.688 \pm 0.029}$ | $0.480 \pm 0.012$ |
| 10 | $0.567 \pm 0.015$ | $\mathbf{0.720 \pm 0.016}$ | $0.465 \pm 0.010$ |
| 15 | $0.577 \pm 0.007$ | $\mathbf{0.739 \pm 0.009}$ | $0.475 \pm 0.009$ |
| 20 | $0.584 \pm 0.013$ | $\mathbf{0.735 \pm 0.011}$ | $0.508 \pm 0.009$ |
| 40 | $0.608 \pm 0.009$ | $\mathbf{0.744 \pm 0.007}$ | $0.536 \pm 0.005$ |
| 80 | $0.619 \pm 0.007$ | $\mathbf{0.744 \pm 0.007}$ | $0.555 \pm 0.009$ |

**Observation:** Both the linear Laplacian and the 1-layer (non-iterative) Lap+PE configurations substantially reduce performance. The full in-context Transformer benefits from iterative Laplacian updates and non-linear kernels, which yield improved manifold alignment and label propagation.

## G  TRANSFORMER BASELINE ARCHITECTURE

**Model:** SmallScaleTransformer (adapted from von Oswald et al., 2023). This serves as an *in-context Transformer pipeline*, where raw features and tokenized labels are jointly embedded, processed by a Transformer encoder, and predictions are made in-context.

**Pipeline Overview:**

- **Input construction:** Raw features $[B, N, D]$ and label IDs $[B, N]$ are embedded via input_projection(raw) + label_embedding(labels) $\rightarrow [B, N, 240]$. Context tokens use true labels; query tokens use a special unknown_label_id.
- **Encoder:** Two Transformer layers (4 heads, $d_{\text{model}}$=240). Each layer uses Multi-Head Self-Attention and a feed-forward block (240→960→240) with LayerNorm and residual connections.
- **Output heads:** Feature head – Linear(240→4) (positional encoding)   Classification head – Linear(4→2) (binary prediction)
- **Training:** Loss computed only on query tokens; context tokens serve as demonstrations. Cross-entropy on query logits vs. true labels; all parameters optimized end-to-end.

**Total Parameters:** 1,442,178 ( 1.4M). This baseline acts as a standard in-context Transformer for comparison with our structured semi-supervised version.

## H  MODEL PARAMETER COUNTS

The parameter counts reported below refer to the total number of **trainable model parameters**—that is, all learnable weights (including attention projections, feed-forward layers, and embedding matrices) within each model configuration. These values indicate the relative model capacity rather than computational cost, and allow for fair comparison between our proposed in-context semi-supervised Transformer and standard baselines.

Table 7: Parameter counts and architectural summary of all compared models.

| Model | Parameters | Architecture Configuration |
|---|---|---|
| ORIG + LR | 627 | Linear logistic regression |
| ORIG + RBF-LR | 628 | RBF kernel logistic regression |
| EIG + LR | 627 | Regression on eigenfeatures |
| ORIG + ICL | 576 | **Our in-context learning Transformer** (on raw inputs) |
| EIG + ICL | 576 | **Our in-context learning Transformer** (on eigenfeatures) |
| ORIG + Transformer | 1,442,178 | **In-context Transformer baseline** |
| ORIG + E2E-ICL (ours) | 10,852 | Lap (2,592) + PE (7,684) + ICL head (576) |

**Parameter Composition (ours):**

- Lap module – 2,592 parameters
- PE module – 7,684 parameters
- ICL Transformer – 576 parameters (in-context classification head)

**Remark 6.** *Unless otherwise stated, all ICL-based models use a single-layer ICL Transformer with the RBF kernel as the core component of the GD ICL head. For all experiments, we vary the labeled-data ratio among a fixed $n = 100$ total samples.*

## I ADDITIONAL RELATED WORK

In the following, we describe several recent work which investigate the use of unlabelled data for in-context learning. Agarwal et al. (2024) show that scaling to many-shot prompts yields large gains across diverse tasks, and further introduce Reinforced ICL (using model-generated rationales) and Unsupervised ICL (inputs only) to reduce dependence on curated labels. Chen et al. (2025) improve many-shot ICL by influence-selecting unlabeled examples, pseudo-labeling them with an LLM, and adaptively choosing demonstrations per query. Li et al. (2025) prove that a single-layer linear-attention model cannot exploit unlabeled data, while depth/looping constructs polynomial estimators akin to EM; they demonstrate looping gains on tabular tasks.

These results motivate our focus on how transformers leverage unlabeled context: rather than adding (pseudo-)labeled shots, we extract structure directly from unlabeled tokens by computing a representation in-context and then using additional attention layers to implement a gradient-based learning algorithm. Our approach complements many-shot/pseudo-labeling by providing an explicit forward-pass mechanism that explains why depth helps in SS-ICL, and is supported by cross-domain evidence—from synthetic manifolds to diffusion images and ImageNet100 features—that full transformers learn geometry-aligned representations consistent with this mechanism.

