# OpenReview forum: "In Context Semi-Supervised Learning"
_ICLR.cc/2026/Conference — ICLR 2026 Poster_

### Official Review · Reviewer_w9xM · 2025-10-28

**Soundness:** 3
**Presentation:** 3
**Contribution:** 2
**Rating:** 4
**Confidence:** 3

**Summary:**

This paper studies semisupervised in-context learning via two-stage Transformer framework. The first stage is to implement representation learning via computing discrete Laplacian and Eigenmaps. While the second is for in-context learning with additional prediction head for classification problems. Experimental results have shown that this two stage framework achieves the best performance compared to the others.

**Strengths:**

1. This paper focuses on a realistic problem, semi-supervised learning, where a large amount of data may lack labels.
2. This paper proposed method that assigned different functional roles to different Transformer blocks, eg, representation extraction and ICL+categorial mapping. Such separation may provide insights into understanding the roles of different Transformer layers.

**Weaknesses:**

1. In Line115, the authors claimed "we are unaware of any prior semi-supervised implementations." However, there are numerous prior works that have explored semi-supervision ICL. For examples,
> [Agarwal et al. 2024] Many-shot in-context learning
>
> [Li et al., 2025) When and How Unlabeled Data Provably Improve In-Context Learning
>
> [Chen et al., 2025] Maple: Many-shot adaptive pseudo-labeling for in-context learning.

while there are many more relevant works. Therefore, I am surprised that the authors are unaware of any prior work on this topic.

2. In Lines 200-212, the authors construct the input $X$ from original dimension $d\times n$ to $(d+n)\times n$ where $d$ is the input feature dimension and $n$ is the sequence length. It is impractical and raises several issues:
- In sequence models, the input length $n$ is typically changing over time;
- It is computationally inefficient, especially when $n\gg d$. However, modern large models are typically capacity of long input sequences, and $n\gg d$ is common in practice.
- According to Appendix A.1, due to such input construction, the number of model parameters have increased from $d\times d$ to $n\times n$, which greatly increases the computational and memory cost.
3. Some notations lack clarity. For examples, what are the dimensions of $\phi^{(i)}$ in Line 233, $\varphi(\phi)$ and $A$ in Line 237?
4. Due to the Weaknesses 2, the authors could only test on the fixed in-context sample size $n=100$.
5. While the authors have shown that OROIG+E2E-ICL achieves the best performance, this method require more computation and larger model parameters to train. Such comparisons may be unfair.

**Questions:**

1. What is $f$ in Eq(1)? It seems that the representation of each raw feature $x^{(i)}$ is independent of others $x$'s, which contradicts the nature of attention and ICL, where representation $f$ at token $i$ should depend on prior examples $(x^{(j)},y^{(j)})_{j<i}$. Otherwise, it is not aligned with standard ICL.
2. Why in Eq(4), the loss is calculated on unlabeled data given that $y^{(i)}$ is unknown? Also, could the authors clarify on this and the Line 175 "When training, this loss is fit to the n−m unlabeled samples (for which we have labels when training)."?
3. Could the author provide more details on the meaning of $w_y-\mathbb{E}(w|f_\ell)$ in Eq (6)?
4. In Eq (7), the left contains an expectation symbol but its expression does not include any expectation operation. Could you clarify on this?
5. Is the $\gamma$ in Eq  (7) equal to the parameter set {$w_c,c=1,2,\cdots C$}?
6. What does $\ell$ represent in Section 3.2?
7. Since E2E is treated as feature exactor, have the authors considered replacing it with other feature extraction methods, such as using pretrained DNN models?

---

> ### Author Response · Authors · 2025-11-17
> **Author Response Part 1/2**
>
> We thank you for the careful and constructive review, and for acknowledging the value in the problem we study. We appreciate the opportunity to address your questions and the weaknesses you identified, which we believe will help us significantly improve the paper.
>
> ---
>
> ### Response to Weaknesses
>
> **W1: On the awareness of prior semi-supervised ICL implementations**
>
> We thank you for pointing out these relevant papers, and we agree that our paper would be strengthened by a more thorough discussion of this body related work. We present a detailed discussion of these papers in **Appendix I** of the current revised draft. We will merge this into the related works section in the next draft. Below we give a brief summary.
>
> * **Agarwal et al. (2024) & Chen et al. (2025, MAPLE):** show that many‑shot prompts can unlock strong performance (including “unsupervised ICL” using inputs only) across diverse tasks in LLMs. These paper show that many‑shot prompts can unlock strong performance across diverse tasks in LLMs, demonstrating empirically that unlabelled data can greatly enhance in-context learning. Our work is complementary: we aim to provide a **mechanistic and algorithmic understanding** of **how Transformers can perform IC-SSL**, starting from first principles (Laplacian eigenmaps, functional GD) and building a specific, interpretable architecture. This contrasts with empirical studies of emergent abilities in pre-trained LLMs, which, as you note, is a realistic and important problem.
>
> * **Li et al. (2025):** We were not aware of Li et al. (2025) at the time of submission, and we thank you for bringing this very interesting work to our attention. This paper provides an elegant, provable analysis of IC-SSL. However, its focus is on a specific, albeit informative, setting of a **binary Gaussian Mixture Model**. Our work aims to provide a framework for a more general class of problems based on the **manifold hypothesis**, as demonstrated in our more diverse synthetic and ImageNet-based experiments. We also note that this paper is associated with NeurIPS 2025 (which publishes after July 24, 2025), and is contemporaneous with our paper. We have however added a proper reference and discussion of this paper in Appendix I as well.
>
> We believe the concurrent emergence of these papers strengthens the case for the community's interest in this problem.
>
> **W2: On practicality, computational/memory cost, and parameter count**
>
> We believe there may be a misunderstanding regarding the complexity and parameter count, which we are happy to clarify. We apologize for any lack of clarity in Appendix A.1 that led to this conclusion.
>
> * **Parameter Complexity:**
> * Our constructions (Lemma 1 and 2) have parameter counts that **do not increase with n**. Though the "input dimension" is indeed $d+n$, the $n\times n$ sub-matrices of the $W^V, W^Q, W^P$ parameters are in fact **scalar multiples of identity** (laplacian layers) or of dimension $k\times k$ (eigenmap layers), where $k$ is desired dimension of learned embedding and is chosen independent of $n$. We added **Remark 1** and **Remark 2** in the current revised draft to emphasize this.
> * In our actual implementation, we impose significant sparsity constraints on the $n\times n$ submatrices of $W^V, W^Q, W^P$ matrices. Specifically, we parameterize them as learnable diagonal matrices. Thus the model's parameter count **does not scale quadratically with sequence length $n$**. **Table 7** in shows the total parameter count of our model (10,852), which is only slightly larger than a single $100\times 100$ matrix.
>
> * **Computational Complexity:** Our model does scale as $O(n^2)$ in computational complexity. However, this $O(n^2)$ scaling is **inherent to the attention mechanism** in *any* standard Transformer, including our baselines. It is not a new parametric complexity. Modern LLMs use efficient attention variants (such as block-diagonal QKV matrices). It is beyond the scope of this paper, but would be an interesting future direction to analyze how our model can incorporate these ideas.
> * **Fixed Context Size:** Our choice of $n=100$ was for consistent experimental comparison across different methods, not a fundamental limit of our model.
>
> We will revise Appendix A.1 to clarify this this distinction between computational and parametric complexity. Our model is very parameter efficient, compared to the Transformer baseline (Table 7). It is this specific, constrained *architecture* (our inductive bias) that allows it to outperform the baseline in low-label regimes, rather than a simple increase in parameters.
>
> ---
> We respond to your other questions in part 2/2

---

> ### Author Response · Authors · 2025-11-17
> **Author Response Part 2/2**
>
> Here are our responses to your questions:
>
> 1.  **On $f(\phi(x^{(i)}))$ in Eq (1) and ICL:**
>     You are correct that $f(\phi(x^{(i)}))$ in Eq (1) appears to be context-independent. This equation, along with Eq (2), is part of the **"Background: In-Context Supervised Learning"** section (Sec 2), where we describe *prior work* that our paper builds upon. In that prior supervised work, the representation $\phi(x^{(i)})$ did not typically depend on the other context points.
>
>     Our **novel contribution** for In-Context Semi-Supervised Learning (IC-SSL) is introduced in the next paragraph, "In-context *semi-supervised* learning." Just before Eq (3), we explicitly define $\phi(X)$ as "a function of **all** observed covariates," meaning the "features $\phi^{(i)}$ are not only dependent on $x^{(i)}$, but also account for context." In our model, $f$ is the classifier (the ICL head), which operates on these **context-dependent features** $\phi^{(i)}$. These $\phi^{(i)}$ are the *output* of our $TF_{rep}$ module (Sec 3.1), which computes the eigenmap from *all* $n$ input tokens. Therefore, the input to $f$ *is* fully context-dependent, which is how our model performs ICL.
>
> 2.  **On Loss (Eq 4) and Line 175:**
>     This is the standard paradigm for **training** a model on a semi-supervised task.
>     * During the **training phase**, we (the researchers) possess the true labels for all $n$ samples.
>     * We *simulate* the semi-supervised setting by *hiding* $n-m$ labels from the model's inference mechanism. Specifically, these $n-m$ labels are *not* used in the functional gradient descent update of $TF_{sup}$ (the sum in Eq. 6 only goes up to $m$).
>     * The loss in Eq (4) is then computed by comparing the model's predictions for these $n-m$ "held-out" points against their **true, known labels**. This is what Line 175 means: "When training... we have labels [for the unlabeled samples]." This is how the model learns to perform the task. At **test time**, these labels are genuinely unknown.
>
> 3.  **On the meaning of $w_{y} - \mathbb{E}(w|f_\ell)$ in Eq (6):**
>     This term is the **functional gradient**, or the "error signal," for the in-context gradient descent step.
>     * **$w_{y^{(j)}}$**: This is the learned, $d'$-dimensional embedding vector for the **observed, true label** $y^{(j)}$ of a *labeled* context point ($j \le m$). It acts as the "target" representation.
>     * **$\mathbb{E}(w|f_\ell^{(j)})$**: This is the **model's predicted expected embedding** at iteration $\ell$. As defined in Eq (7), it is the average of *all* possible class embeddings $\{w_c\}$, weighted by the model's *current* predicted probabilities for each class (i.e., the softmax output given $f_\ell^{(j)}$).
>     The difference between the "target" ($w_{y^{(j)}}$) and the "prediction" ($\mathbb{E}(w|f_\ell^{(j)})$) is the error that the GD step (Eq. 6) seeks to correct, thereby updating the latent function $f$.
>
> 4.  **On the expectation in Eq (7):**
>     The notation $\mathbb{E}(w|f_\ell^{(j)})$ on the left is shorthand. The expression on the right-hand side **is** the explicit definition of this expectation. It represents the expected value of the class embedding vector $w$, conditioned on the current latent function $f_\ell^{(j)}$.
>     * It is an expectation $E_{c \sim P(\cdot \mid f_\ell^{(j)})}[w_c]$.
>     * This expands to $\sum_{c=1}^C w_c \cdot P(c | f_\ell^{(j)})$.
>     * The term in the brackets, $\frac{\exp[w_c^Tf_\ell^{(j)}]}{\sum_{c^\prime=1}^C \exp[w_{c^\prime}^Tf_\ell^{(j)}]}$, *is* that probability $P(c | f_\ell^{(j)})$, calculated via the standard softmax function.
>
> 5.  **On $\gamma$ in Eq (7):**
>     The $\gamma$ refers to the **learnable parameters of the MLP** ($\text{MLP}_\gamma(\cdot)$) that follows the attention layer in our ICL Transformer. As stated in the text and detailed in Appendix C, we use this MLP to *approximate* the non-linear expectation function. $\gamma$ and $\{w_c\}$ are distinct sets of learnable parameters.
>
> 6.  **On $\ell$ in Section 3.2:**
>     $\ell$ represents the **iteration step of the functional gradient descent (FGD)** algorithm. Our construction is designed to *implement* this iterative algorithm in its forward pass. We explicitly **map each GD step $\ell$ to a Transformer layer $\ell$** in the $TF_{sup}$ module. Therefore, an $L$-layer $TF_{sup}$ module performs $L$ steps of this in-context functional gradient descent.
>
> 7.  **On replacing $TF_{rep}$ with pretrained DNNs:**
>     This is an **excellent point**, and our ImageNet100 experiments studies exactly this. In that section, the *input* $x^{(i)}$ to all models (ours and baselines) is not raw pixels, but VGG features (i.e. pretrained DNN output). Figure 3 shows that our end-to-end model outperforms a Transformer baseline, even though both take as input the VGG features. This demonstrates that our proposed **in-context feature learning mechanism yields benefits even when given pretrained DNN features instead of raw pixels**.

---

> ### Author Response · Authors · 2025-11-25
>
> Dear reviewer w9xM, it came to our attention that OpenReview may not have sent notifications for our response above due to visibility settings. We are posting this note to serve as notification of our response. Please disregard this if you are already aware of our rebuttal. Thank you again for taking the time to review our paper.

---

### Official Review · Reviewer_NFR7 · 2025-10-31

**Soundness:** 3
**Presentation:** 4
**Contribution:** 3
**Rating:** 4
**Confidence:** 4

**Summary:**

This paper introduces IC-SSL, a novel Transformer architecture explicitly constructed to perform in-context semi-supervised learning. The core contribution is a two-stage model: 1) A representation-learning stage ($TF^{\phi}$) that is designed to leverage unlabeled data by computing Laplacian Eigenmaps, a classic spectral method for manifold learning. 2) An ICL stage ($TF_{sup}$) that uses these learned representations, along with the few available labels, to perform classification by provably mimicking functional gradient descent. The authors provide theoretical analysis (in the appendices) to demonstrate that their specialized attention modules (e.g., $Attn_{rbf}$, $Attn_{linear}$) can indeed be parameterized to execute these specific algorithmic steps. Empirical results on synthetic manifolds and real-world data demonstrate the method's superiority over baselines, especially in low-label regimes.

**Strengths:**

1. **Theoretical Quality**: The appendices provide a rigorous "proof-by-construction." The mathematical links between specific attention mechanisms and algorithmic steps (like $Attn_{linear}$ for power iteration) are non-trivial and well-executed.

2. **Empirical Effectiveness**: The method is shown to be highly effective, outperforming strong baselines, which validates the authors' algorithmic choices.

**Weaknesses:**

1. **Lack of Justification for Algorithmic Complexity**: The paper jumps directly to a sophisticated, iterative, non-linear SSL algorithm (Laplacian Eigenmaps). It doesn't first establish why this complexity is needed. The theoretical argument would be much stronger if it included an analysis (or even a citation to existing work) demonstrating that simpler, non-iterative (e.g., single-layer) or linear models fundamentally fail at this task.

2. **Specialization of the Spectral Method**: The choice of the Laplacian $\hat{\mathcal{L}}$ (via $Attn_{rbf}$) is a very specific one, tailored to non-linear manifolds. This is a strong design choice, but it's presented without robustly comparing it to simpler alternatives. A major question is whether a linear spectral method (e.g., PCA) would be sufficient. Given that the $TF^{\phi}$ module itself uses $Attn_{linear}$  (which is known to be related to matrix power computation), a standard model might more naturally learn to compute powers of the data covariance matrix ($X^\top X$) rather than the complex $\hat{\mathcal{L}}$. The paper fails to discuss the trade-offs of its specific, non-linear choice.

**Questions:**

1. **On the Necessity of Iteration**: Your model relies on iterative algorithms (power iteration, GD) implemented via depth/loops. What are the theoretical trade-offs of this explicit iteration versus a hypothetical non-iterative (e.g., single-layer) but perhaps very wide model? Is there a theoretical justification for why an iterative approach is fundamentally necessary for IC-SSL, thus motivating your design?

2. **On the Choice of Spectral Method (Linear vs. Non-linear)**: Your $TF^{\phi}$ module is complex: it first uses $Attn_{rbf}$ to compute the Laplacian $\hat{\mathcal{L}}$, and then uses $Attn_{linear}$ to compute its eigenmap via power iteration. A simpler approach might skip the first step. Given that stacked $Attn_{linear}$ layers are mathematically known to compute matrix powers, a standard Transformer might implicitly learn to compute powers of the data covariance matrix ($X^\top X$), a linear spectral method (i.e., PCA).
  * (a) Can you theoretically justify the necessity of your explicit, two-step, non-linear spectral method ($\hat{\mathcal{L}}$) over this simpler, linear spectral alternative ($X^\top X$)?
  * (b) Did you empirically compare your $TF^{\phi}$ module against a simpler $TF^{\phi}$ baseline that only uses stacked $Attn_{linear}$ to (implicitly) learn a linear spectral embedding?

3. **On the Algorithmic Structure (Decoupled vs. Coupled)**: You chose a 'decoupled' two-stage design (Learn Reps -> Classify), which is highly interpretable. However, many successful SSL algorithms (like EM) are 'coupled,' iteratively refining label-estimates and model parameters. What are the theoretical limitations of your decoupled approach? Did you explore or consider a more coupled, end-to-end model where the $TF^{\phi}$ and $TF_{sup}$ stages are interleaved at each layer?

---

> ### Author Response · Authors · 2025-11-15
> **Author Response 1/2**
>
> We thank the reviewer for the careful reading. We also thank the reviewer for raising a number of astute theoretical and empirical questions, which help shed light on several subtle reasoning made in the paper. We agree it is important to justify *why* this level of algorithmic and architectural complexity is warranted. We also revised our draft to include the reviewer's suggested ablation experiments, which are very helpful for elucidating the various design choices.
>
> We will first address Question 2 and then return to Question 1.
>
> ---
>
> ### 2(a) Theoretical justification of non-linear spectral method
>
> **Why not PCA / covariance powers?**
>
> On the question of why we do not use a linear spectral method such as PCA or something based only on powers of the data covariance matrix, our motivation is both intuitive and theoretical.
>
> Intuitively, real data often lives on **non‑linear manifolds**, and PCA has difficulties dealing with symmetries in the data, which can appear quite naturally. In general, the non‑linearity and increased complexity of the graph Laplacian allow for more flexibility and the ability to handle a variety of scenarios with a simple tuning of the bandwidth parameter. Our first stage is explicitly designed to be manifold‑intrinsic: it uses **local neighborhoods** on the data manifold rather than **global second‑order moments**.
>
> **We present below a simple manifold which is not separable via PCA.**
>
> Consider two classes, denoted by $y=0$ and $y=1$. Each class is supported on a different concentric circle in $\mathbb{R}^2$, i.e.
> * if class $y=0$, sample $x$ uniformly on the circle of radius $r_1$;
> * if $y=1$, sample $x$ uniformly on the circle of radius $r_2>r_1$.
>
> (This example can be turned into a connected manifold by connecting the circles with a thin line segment; it is cleaner to first consider the concentric case.)
>
> 1. **PCA cannot separate the two classes:** By rotational symmetry, the population mean is zero and the covariance is a scalar multiple of the identity:
>    $
>    \Sigma = \mathbb{E}[xx^\top] = \sigma^2 I_2.
>    $
>    Thus PCA (or any method that only has access to $\Sigma$ and its powers) sees no preferred direction: every orthonormal basis is an eigenbasis, and $\Sigma^k = \sigma^{2k} I_2$ for all k. In the infinite‑data limit, any linear spectral method based solely on the data covariance (including stacks of linear attention layers that implicitly compute matrix powers) cannot distinguish the inner and outer circles; it is fundamentally blind to the “radius‑only” label structure because the second‑order statistics of the two classes coincide.
>
> 2. **Linear decision boundaries are then stuck.**
>    In such a representation, any classifier implemented as a single linear decision boundary in $\mathbb{R}^2$ must cut across both circles and therefore incurs irreducible classification error (one can show that every affine separator misclassifies a constant fraction of points).
>
> By contrast, if one applies a graph Laplacian approach to the *same* data, the situation changes. With an RBF kernel and a bandwidth chosen so that intra‑circle edges are strong and inter‑circle edges are weak, the induced similarity graph is essentially two disconnected (or very weakly connected) components. **The second eigenvector of the graph Laplacian is then (up to normalization) an indicator of which circle a point belongs to.** In this embedding, a one‑dimensional linear classifier suffices to cleanly separate the two circles.
>
> This is a stylized counterexample, but it is quite close in spirit to the manifolds we study in the paper: our Swiss roll, flat torus, and sphere examples all exhibit symmetries (e.g., “inner vs. outer loop” in the Swiss roll) of exactly this type. The key phenomenon is:
>
> * **Linear spectral methods based on ($X^\top X$)** struggle when the population covariance is close to isotropic and the label structure depends on nonlinear manifold coordinates (such as radius).
> * **The graph Laplacian**, by construction, is sensitive to local geometry and approximates the Laplace–Beltrami operator as we refine the graph and add more samples, giving a manifold‑intrinsic notion of smoothness and a principled way to separate such structures.
>
> There are absolutely interesting domains where PCA works very well, but in settings like this example and our synthetic manifolds, symmetries prove difficult for purely linear spectral approaches, whereas Laplacian‑based embeddings handle them more gracefully. This is why we view the Laplacian as a principled choice informed by the manifold hypothesis.
>
> Finally, we remark that the $Attn_{rbf}$ (see (9) in paper) is closely related to the standard softmax-activated Transformer, and are mathematically equivalent if input tokens are normalized to the unit sphere, e.g. via LayerNorm.
>
> (continued below)

---

> ### Author Response · Authors · 2025-11-15
> **Author Response Part 2/2**
>
> ### 2(b) Empirical comparison of non-linear vs linear spectral methods
>
> We have run additional experiments on our synthetic product manifold where we remove $Attn_{rbf}$ and use only stacked $Attn_{linear}$ layers, keeping all other aspects of the model as close as possible. We present the full results in **Table 6, Appendix F.8** of the current revised draft. For ease of reference, we include a subset of results below:
>
> | Context Size | Our Model | Only stacked $Attn_{linear}$| Non-Iterative|
> | :------- | :------: | -------: | -------: |
> | 3| 0.66 | 0.48 |0.53|
> | 20   | 0.72   | 0.47   | 0.58 |
> | 80   | 0.74  | 0.56   | 0.62|
>
> Comparing the first two columns, we see that **$Attn_{rbf}$ achieves higher accuracy than pure $Attn_{linear}$ in all context size regimes.**
>
> We also want to reiterate that our module is trained end‑to‑end, and the model is fundamentally allowed to learn a different embedding. Empirically, it does *not* simply recover the graph Laplacian eigenmaps; it *outperforms* the variant where we precompute the Laplacian and then use the same ICL layers (see e.g. Figure 4, blue vs green line). We start from a theoretically motivated Laplacian‑based construction, but during training we allow it to deviate in ways that appear helpful.
>
> ---
>
> ### 1. On the necessity of iteration
>
> Returning now to Question 1, on the necessity of iteration.
>
> Conditioning on the idea that it is worthwhile to pursue Laplacian eigenmaps as a representation (for the reasons above and in part 1), there is a natural motivation for explicit iteration in our architecture: All algorithms that we know of for computing eigenmaps/functional gradient descent (ICL head) require some form of iteration (power iteration, subspace iteration, gradient descent, etc.). However, we are *not aware of any formal lowers bound that rule out all non‑iterative algorithms*.
>
> Our representation stage therefore uses a power‑iteration‑like block, and our ICL head fits the classifier by gradient descent on a Gram matrix in‑context, which is also an iterative algorithm. In the forward pass of a Transformer, these iterations are realized via depth/loops.
>
> It is conceivable that an extremely wide, non-iterative model may be able to simultaneously approximate multiple steps of some iterative algorithm, but might trade away algorithmic clarity and sample efficiency for a more opaque parameterization. In contrast, our design makes the algorithmic structure explicit in depth/loops. Given that the algorithms we want to approximate (power iteration and gradient descent) are seemingly inherently iterative, this is our theoretical justification for the necessity of iteration in IC‑SSL.
>
> We also conduct an ablation study comparing our model versus a variant where we use a single non-looped layer to learn in-context representation (**Table 6, Appendix F.8** of our current revised draft). An subset of results are presented in last column of the table above. In all regimes, **depth/loops lead to a significantly higher accuracy**.
>
> ---
>
> ### 3. On the decoupled vs. coupled algorithmic structure
>
> Finally, on Question 3 about the algorithmic structure (decoupled vs. coupled):
>
> We agree that the idea of a coupled, interleaved design is very interesting, and we thank the reviewer for raising this possibility. Our current choice of a decoupled two‑stage model (learn representations → classify) was driven by interpretability and by the desire to give each stage a clear algorithmic role: the first approximates Laplacian eigenmaps; the second performs in‑context gradient descent.
>
> At the same time, because the model is trained end‑to‑end, the stages are not rigidly separated. The representation module is **not forced to compute exactly the graph Laplacian eigenmap, and the ICL head can further refine and exploit whatever representations the first stage produces**. In that sense, there is already some “coupling” through joint optimization: the effective purpose of each stage can bleed into the other.
>
> We do not see a fundamental theoretical limitation of a decoupled approach in our setting; rather, we view a coupled, interleaved architecture as a non‑trivial and promising extension. Designing an architecture that passes information back and forth in an interleaved manner *and* retains the algorithmic guarantees we currently have is an exciting and substantial future work—directly inspired by your comment.
>
> ---
>
> ### Closing
>
> Thank you again for the thoughtful and theoretically oriented review. In the revision, we plan to integrate these very good points you brought up and introduce these theoretical arguments and experimental results here. We hope this makes clear that our design choices are not ad hoc, but are motivated by the manifold hypothesis, by the theory of graph Laplacians and Laplace–Beltrami operators, and by the need to handle complex geometries in a principled way.

---

> > ### Comment · Reviewer_NFR7 · 2025-11-23
> > **Nice Rebuttal. I Really Appreciate It.**
> >
> > I thank the authors for their detailed response and the inclusion of the additional ablation study.
> >
> > I particularly appreciate the "concentric circles" counter-example provided in response to Question 2. This example elegantly clarifies the theoretical boundary between your proposed method and simpler linear alternatives (such as implicit power iteration on $X^\top X$). It convincingly demonstrates that while standard linear attention strategies might suffice for tasks governed by linear separability (like GMMs), your explicit construction of a non-linear manifold-intrinsic operator (the Laplacian) is theoretically necessary for handling geometric symmetries where global second-order statistics are isotropic. This successfully addresses my primary concern regarding the justification for the architectural complexity and highlights the unique value of your constructive approach.In light of this strong theoretical defense and the empirical validation, I am raising my score to 6.
> >
> > However, I also note the valid points raised by Reviewer w9xM, particularly regarding the discussion of related/concurrent works and clarifications on model complexity. I will be closely monitoring the ongoing discussion between the authors and Reviewer w9xM. If those concerns are resolved satisfactorily, **I am open to further increasing my score to 8.**

---

> > > ### Author Response · Authors · 2025-11-23
> > >
> > > We sincerely thank Reviewer NFR7 for recognizing our theoretical contributions and for raising their score, as well as for the reminder about notifications.
> > >
> > > We hope our subsequent discussion with Reviewer w9xM will address your concerns. We will also be happy to answer any additional questions you may have about related/concurrent works or model complexity.
> > >
> > > Finally, thank you again for the valuable discussions and review.

---

### Official Review · Reviewer_TXiC · 2025-11-01

**Soundness:** 3
**Presentation:** 3
**Contribution:** 3
**Rating:** 6
**Confidence:** 2

**Summary:**

The paper studies a setting of semi-supervised in-context learning (termed IC-SSL), where the input consists of many examples, a few of which are accompanied by gold labels. From a theory of ICL perspective, the authors construct and train small transformers on various classification tasks with synthetic data drawn from manifolds, and on one more realistic task using interpolations between images in ImageNet. They propose a specific model for this setting that outperforms baseline transformers, based on an argument about how semi-supervised learning might be performed in context mechanistically.

**Strengths:**

S1. The paper thoroughly explores semi-supervised ICL, presenting experiments on carefully designed synthetic data, using multiple random seeds, and reporting interesting ablations.

S2. The comparison of performance on in-domain and out-of-domain data when training on data generated from different subsets of the classification manifolds was quite interesting.

S3. The proposed E2E-ICL is both mathematically well-justified and empirically effective at semi-supervised ICL.

**Weaknesses:**

W1. It would be nice to at least mention recent [empirical work](https://arxiv.org/abs/2404.11018) on unsupervised ICL.

W2. The claim in line 343 that the model is competitive on real data seems a bit ill-supported, as even the ImageNet task is a fairly synthetic classification task.

**Questions:**

Q1. Can you elaborate on line 72, the claim that the model "provably" performs ICL?

Q2. In Figure 1, you add the labels for the few labeled examples back into the representation in step B. Why not allow representations to be learned with awareness of these labels?

Q3. What is the transformer baseline in Figure 3/10? It's not in the list of models in 4.1.

Q4. Can you provide the parameter counts for all model types trained? It's a bit hard to tell from the descriptions alone whether some methods might have more learnable expressivity.

My work is more in the empirical direction, so some of these clarifications may be sufficiently clear already in the text for a more theory audience.

Other comments:
- Please fix citations to use \citep{} where appropriate, especially in the introduction.
- for the sentence on line 31 about functional gradient descent, would be nice to cite back to the prior work in this area
- I found the contributions section 1.1 to be quite long and a bit confusing before reading the rest of the paper; I think this might be better positioned near the end of the work, with just a short summary of contributions in the introduction.
- 4.1 and F7 are very similar text; F7 could perhaps be merged into 4.1.
- it would be nice if the line in Fig 4(b) that's a directly comparable result from 4(a) was the same color in both, though I understand the color scheme is meant to indicate the different model types.

---

> ### Author Response · Authors · 2025-11-15
> **Author Response Part 1/2**
>
> We thank you for the careful and thoughtful review, and for the positive assessment of the paper’s good soundness, presentation, and contributions. We especially appreciate your comments on the experimental design and the proposed E2E‑ICL mechanism. Below we address your weaknesses and questions in turn and describe concrete changes we will make in the revised manuscript.
>
> ---
>
> ### W1. Relation to recent empirical work on unsupervised / many‑shot ICL
>
> We thank the reviewer for bringing to our attention this very relevant study by Agarwal et al. We have added a short “Additional Related Work” entry (**Appendix I**) covering **Many‑Shot ICL** (Agarwal et al., 2024), and compares their setting with ours. This direction of work indeed served as an inspiration to our more theoretical work. To briefly summarize: Agarwal et al. show that *many‑shot* prompts can unlock strong performance (including “unsupervised ICL” using inputs only) across diverse tasks in LLMs, highlighting emergent behavior when labels are absent or minimal.
>
> Their empirical results motivate our mechanistic study: rather than adding (pseudo‑)labels or relying on emergent LLM behaviors, we **explicitly construct** an IC‑SSL mechanism that (i) builds a geometry‑aware representation from *unlabeled* tokens via a Laplacian‑eigenmap stage and (ii) performs *in‑context* gradient‑based classification on top (Sec. 3). We provide a more detailed discussion in Appendix I.
>
> ---
>
> ### W2. “Competitive on real data” vs synthetic / artificial tasks
>
> Thank you for flagging this; we agree the current phrasing is too strong and also mixes terminology. Our intent was to distinguish between:
>
> * **Synthetic data**: data that are *algorithmically generated*
> * **Artificial tasks on real data**: tasks where the *data* are real (e.g., ImageNet images / VGG features), but the *task* and context construction are artificial and not motivated by a real world problem.
>
> In that sense, our ImageNet100 and Stable‑Diffusion experiments are closer to “artificial tasks on real data” than to fully synthetic contexts. Prior theory‑style ICL work we build on (e.g., linear/RKHS regression and categorical ICL) focused on fully synthetic data, with the exception of one recent Wang et al 2025 paper that also considered ImageNet data for synthetic algorithmic tasks.
>
> We agree that the sentence on line 343 currently overstates this and can be misleading. In the current revision, we have updated the description from
> > “our two-stage model is competitive on real data…”
>
> to
>
> > “our two-stage model is competitive on on ImageNet‑based artificial IC‑SSL tasks, using real image features..”
>
> In the next draft, we will explicitly note state in the Limitations section that our experiments are *artificial semi‑supervised ICL tasks* that serve as an intermediate step between fully synthetic algorithmic ICL and more realistic downstream tasks.
>
> (continued below)

---

> ### Author Response · Authors · 2025-11-15
> **Author Response 2/2**
>
> ### Q1. “Provably” performs ICL (line 72)
> We thank the reviewer for identifying the imprecise description. What we mean is the following:
>
> * In Appendix A and Section 3.2, we exhibit an explicit **construction** of a Transformer whose self‑attention + MLP blocks implement a finite number of steps of **functional gradient descent** on the in‑context cross‑entropy loss for the categorical model in Eq. (1), under an RKHS assumption on (f).
> * Concretely, we derive the functional GD iteration (6)
>   $$
>   f_{\ell+1}^{(i)} = f_\ell^{(i)} + \sum_{j=1}^m \alpha \bigl[w_{y^{(j)}}-\mathbb{E}(w|f_\ell^{(j)})\bigr],\kappa(\phi^{(i)},\phi^{(j)}),
>   $$
>   that suffices to solve ICL. We then show (in Appendix B+C) that a Transformer layer with appropriately chosen parameters can implement the update (6) in its forward pass.
> * Thus “provably” refers to the existence of a parameter setting where the Transformer implements the desired ICL algorithm for categorical observations.
>
> To avoid overclaiming, we have **rephrased line 72** to:
>
> > “The second stage is a construction for a Transformer‑based ICL module that **can be shown to perform in‑context learning for categorical observations by implementing functional gradient descent in its forward pass** (Appendix B and C).”
>
> ---
>
> ### Q2. Why Stage B adds labels only after representation learning?
>
> This was a deliberate design choice, and we agree it is worth explaining more clearly.
>
> 1. **Interpretability / mechanistic analysis.**
>    By separating the modules so that $TF_{\text{rep}}$ never sees labels, we obtain a representation learner whose behavior we can compare directly to both:
>
>    * The ground‑truth Laplacian eigenmaps, and
>    * A scale‑matched baseline Transformer trained end‑to‑end.
>      This makes it easier to ask “what is the baseline actually doing?” and to align its learned geometry with our constructive reference (e.g., via mNN / alignment metrics in Table 2).
>
> 2. **Generality across label ratios.**
>    Our IC‑SSL setting spans very low labeled ratios (e.g. 3 labeled points out of 100). Having a representation module that depends only on the unlabeled + covariate structure makes it naturally **robust to changing label availability**.
>
> We do fully agree this suggestion is an interesting and promising future direction that can significantly improve the prediction accuracy. We finally remark that in practice we train the full two‑stage model end‑to‑end, so information can “bleed” between modules: although the theoretical separation enables a clean analysis of the representation, the implemented model can still adapt its features using label information through shared gradients.
>
> ---
>
> ### Q3 & Q4. Transformer baseline and parameter counts
>
> **Q3 (baseline definition).**
> The “Transformer baseline” in Figures 3/10 is a **scale‑matched encoder‑only Transformer** with:
> * Standard multi‑head self‑attention and MLP blocks,
> * Inputs consisting of the same covariates, with labels (for the labeled subset) projected into the embedding space and unlabeled points tagged with a special “unknown” token, following MetaICL‑style setups.
> Training uses cross‑entropy on query tokens only; **total parameters ≈ 1.44 M**. We use the same end-to-end in‑context training protocol across methods. We provide details in Appendix G of the current revised draft.
>
> **Q4 (parameter counts / expressivity).**
>
> **Appendix H** of the current revised draft tabulates parameter counts for all models (Table 7). The **baseline Transformer** has **1,442,178** parameters; our **end‑to‑end IC‑SSL** model uses **10,852** parameters in total (**Laplacian** 2,592 + **Eigenmap/PE** 7,684 + **ICL head** 576). For reference, **ORIG+ICL** and **EIG+ICL** heads each have **576** parameters. These figures are intended to make the capacity comparison explicit and transparent. (**Appendix H**, Table 7).
>
> Crucially, EIG+ICL/LR assume Laplacian eigenfeatures are precomputed offline, whereas our IC-SSL must learn these features in-context; most of the extra parameters therefore substitute for offline spectral computation, not extra classifier capacity. We chose the baseline Transformer by matching the internal dimensions of our model, but we do not impose any sparsity constraints on the baseline Transformer parameter matrices, which leads to many more parameters. We note that our model outperforms the baseline Transformer despite using only a small fraction of the parameters.
>
> ---
>
> ### Other comments
>
> We agree with a large majority of your formatting and presentation suggests and thanks to them we believe we will have a much nicer presentation. Especially with regards to the contributions section.
>
> ---
>
> Once again, we thank the reviewer for their very constructive and clear feedback. Your suggestions on related work, wording, architectural clarification, and presentation will substantially improve the final version of the paper.

---

> ### Author Response · Authors · 2025-11-25
>
> Dear reviewer TXiC, it came to our attention that OpenReview may not have sent notifications for our response above due to visibility settings. We are posting this note to serve as notification of our response. Please disregard this if you are already aware of our rebuttal. Thank you again for taking the time to review our paper.

---

### Official Review · Reviewer_jojL · 2025-11-01

**Soundness:** 2
**Presentation:** 2
**Contribution:** 2
**Rating:** 4
**Confidence:** 3

**Summary:**

This paper introduces In-Context Semi-Supervised Learning (IC-SSL), extending Transformer-based in-context learning to settings with few labeled and many unlabeled samples. It proposes a two-stage, end-to-end Transformer: the first stage learns representations from unlabeled data by constructing a Laplacian and computing an eigenmap, and the second stage performs in-context supervised learning to infer labels for unlabeled samples. This enables both representation learning and label propagation within a single forward pass. Experiments on synthetic manifolds and images show it outperforms baselines, leveraging unlabeled data to improve performance with few labels, and generalizes well in and out of distribution.

**Strengths:**

Strengths:

1. This paper presents a novel extension of in-context learning to the semi-supervised setting, enabling Transformers to jointly exploit labeled and unlabeled examples within a unified framework.

2. The proposed two-stage architecture—combining Laplacian-based representation learning with an in-context gradient descent head—offers rare mechanistic interpretability and provides insight into how Transformers may implement semi-supervised reasoning.

3. Empirically, the method demonstrates strong label efficiency and outperforms competitive baselines across synthetic manifolds, product manifolds, and ImageNet100, showing promising generalization and transfer, especially under low-label regimes.

**Weaknesses:**

Weaknesses:

1. Despite its contributions, the experimental evaluation is limited in scale and focuses primarily on small episodic settings, leaving open questions about robustness in large, noisy, or highly diverse real-world scenarios.

2. The model relies on a strong geometry-driven inductive bias, which may restrict flexibility when the data does not exhibit clear manifold structure, and the architecture requires non-standard attention variants and customized modules that could hinder practical adoption.

3. Additionally, the overall complexity introduces training sensitivity and may limit applicability beyond the studied domains.

**Questions:**

Please see the weaknesses.

---

> ### Author Response · Authors · 2025-11-13
> **Author Response Part 1/2**
>
> We thank the reviewer for the constructive feedback and comments. Below, we address each of the weakness raised.
>
> ## Experimental Scale and Robustness
> > Despite its contributions, the experimental evaluation is limited in scale and focuses primarily on small episodic settings, leaving open questions about robustness in large, noisy, or highly diverse real-world scenarios
>
> 1. We reiterate that our goal is **mechanistic understanding of in‑context semi‑supervised learning** (IC-SSL), not benchmarking large‑scale performance. To *isolate and study the mechanism*, we deliberately use controlled episodic setups where geometry, label sparsity, and baselines are precisely controlled. Within this design choice, our experiments span a wide range of difficulty and data types:
> * **Synthetic low-dimensional manifolds in ℝ³** (Fig. 2);
> * **Higher-dimensional product manifolds** up to ℝ¹⁵ with more intricate decision boundaries (Fig. 4c);
> * **Image manifolds in pixel space** generated by Stable Diffusion (Fig. 5);
> * **Image manifolds in VGG feature space** (ImageNet100 in VGG) including representational analyses of a standard Transformer baseline (Fig. 3, Table 1).
>
> Across these domains, we **consistently observe that the end‑to‑end Transformer performs IC‑SSL via the mechanism we propose**: geometry-aware representation learning followed by an in‑context gradient-descent–style update.
>
> 2. We agree that many real applications involve much larger, noisier, and more heterogeneous contexts. Our work is intended as a **first mechanistic step** toward understanding such systems. Concurrent work has already shown that large Transformers can perform semi‑/unsupervised ICL in realistic language tasks [1,2], but these papers do not explain *how* this happens internally. Our results complement them by providing a concrete mechanistic account validated on increasingly complex and realistic domains. Extending our analysis to large-scale LMs and more diverse real‑world datasets is an important direction for future work, which we will emphasize more clearly in the revised Limitations section.
>
> 3. Importantly, this evaluation style follows established practice for mechanistic ICL work and has been recognized at top venues. Several influential studies that analyze simpler supervised settings (e.g., linear/logistic regression in Euclidean space) have been published at ICML’23, NeurIPS’23, and JMLR’24 [3–5]. Our contribution extends that line by moving from supervised to semi‑supervised ICL and by further validating the mechanistic account across synthetic manifolds, diffusion‑image manifolds, and ImageNet100 features (Figs. 3–5, Table 1).
>
> ## “Non-standard” attention variants and customized modules
> > …the architecture requires non-standard attention variants and customized modules that could hinder practical adoption.
>
> Implementation‑wise, **our model is a standard Transformer trained end‑to‑end** (see Sec. 3). The differences from a vanilla encoder are:
>
> 1. **Sparsity constraints** on Q/K/V: we impose simple block-diagonal/diagonal patterns that *reduce* the number of learnable parameters and encode permutation/equivariance structure. An unconstrained Transformer can in principle represent the same operations; our constraints are an explicit inductive bias, not a new primitive.
> 2. **Alternative attention kernels** in parts of the representation-learning stage: we use linear attention modules instead of softmax attention. Such “linear attention” variants, though less common than softmax attention, have been extensively studied, and saw effective adoption in large-scale language models [6,7]. The RBF attention in Lemma 1 is less standard, but can be shown to be equal to standard softmax attention under layer normalization.[8]
>
> Thus, our construction remains fully implementable in any standard Transformer stack: the “non‑standard” components are specific parameterizations of existing attention layers, chosen to make the connection to Laplacian eigenmaps and functional gradient descent transparent rather than to introduce a new architecture intended for deployment.
>
> (continued below)

---

> ### Author Response · Authors · 2025-11-19
> **Author Response Part 2/2**
>
> ## Geometry-driven inductive bias and flexibility
>
> > “The model relies on a strong geometry-driven inductive bias, which may restrict flexibility when the data does not exhibit clear manifold structure.”
>
> The geometry-aware inductive bias is **intentional**: it is what allows us to connect the Transformer’s computation to classical manifold-based SSL and to derive a clear mechanistic understanding. Importantly, our experiments suggest that this bias is *aligned* with how unconstrained Transformers behave (Figure 3), rather than being an unrealistic restriction:
>
> * **Performance:** Our construction outperforms a scale‑matched, fully unconstrained Transformer on semi‑supervised image classification tasks, both on diffusion image manifolds and on ImageNet100 (Figs. 3 and 5), despite using fewer parameters.
> * **Internal mechanism:** The middle/right panels of Fig. 3 and Table 1 show that the baseline Transformer learns representations that are strongly aligned with our Laplacian-based features and undergo a similar phase transition in separation score. This indicates that our model is a **useful mechanistic proxy for the behavior of standard Transformers**, not an artificial construction.
>
> Regarding the manifold assumption: while raw inputs may not always lie on an explicit low-dimensional manifold, **learned feature spaces often do**. Our IC‑SSL module operates effectively both on **raw pixels** (diffusion-generated manifolds) and on **high-level VGG features** (ImageNet100). This suggests that the method is **composable** with existing representation-learning backbones, which frequently produce geometrically structured embeddings.
>
> ## Model complexity and training sensitivity
>
> > *“Additionally, the overall complexity introduces training sensitivity and may limit applicability beyond the studied domains.”*
>
> Conceptually, we describe two stages (spectral representation + in‑context GD), but **in practice this is a single Transformer with standard layers** whose parameters are trained jointly via a single IC‑SSL objective (Eq. (4)). The imposed sparsity patterns reduce parameter count and enforce useful symmetries; in our experiments this **improves stability rather than harming it**.
>
> Empirically, we did **not observe training sensitivity issues**:
>
> * The same architecture and nearly identical hyperparameters are used across all manifold, image-manifold, and ImageNet100 experiments.
> * We report mean ± standard deviation over three seeds for all methods; in many plots the error bars for our model are so small that they are barely visible, indicating robust optimization and insensitivity to initialization.
>
> We will clarify these implementation details and explicitly state in the revised text that, across all domains considered, training our model was stable and did not require delicate tuning.
>
> ---
>
> **References**
>
> [1] Agarwal et al., *Many-Shot In-Context Learning*, 2024.
>
> [2] Chen et al., *MAPLE: Many-Shot Adaptive Pseudo-Labeling for In-Context Learning*, 2025.
>
> [3] von Oswald et al., *Transformers Learn In-Context by Gradient Descent*, ICML 2023.
>
> [4] Ahn et al., *Transformers Learn to Implement Preconditioned Gradient Descent for In-Context Learning*, NeurIPS 2023.
>
> [5] Zhang et al., *Trained Transformers Learn Linear Models In-Context*, JMLR 2024.
>
> [6] Katharopoulos et al., *Transformers are RNNs: Fast Autoregressive Transformers with Linear Attention*, ICML 2020.
>
> [7] Team et al, *Kimi Linear: An Expressive, Efficient Attention Architecture*, 2025.
>
> [8] Cheng et al, *Transformers implement functional gradient descent to learn non-linear functions in context*, 2024

---

> ### Author Response · Authors · 2025-11-25
>
> Dear reviewer jojL, it came to our attention that OpenReview may not have sent notifications for our response above due to visibility settings. We are posting this note to serve as notification of our response. Please disregard this if you are already aware of our rebuttal. Thank you again for taking the time to review our paper.

---

### Author Response · Authors · 2025-12-03
**Rebuttal Summary (1/5 Main Contributions and Strengths)**

Below, we provide a concise, self‑contained summary of (1) the main contributions and strengths *as identified by the reviewers* and (2) the main criticisms and our response.

## Summary of contributions and reviewer‑noted strengths

Our paper makes three main contributions:

1. **Formulating in-context semi-supervised learning (IC‑SSL).** We extend Transformer-based in-context learning to a semi-supervised setting, broadening the concept of “context” to leverage both labeled and unlabeled data within a unified end-to-end Transformer architecture.

   The reviewers affirm the **importance** of studying in-context semi-supervised learning:

   > **w9xM**: This paper focuses on a **realistic problem**, semi-supervised learning, where a large amount of data may lack labels.

   > **jojL**: This paper presents a **novel extension of in-context learning to the semi-supervised setting**, enabling Transformers to jointly exploit labeled and unlabeled examples within a unified framework.

2. Theoretically, using *standard Transformer components*, we **construct** a two-stage architecture, geometry-aware representation learning, followed by an in-context gradient-descent–style ICL module, thus providing a clear mechanistic explanation for how Transformers *can* perform in-context semi-supervised learning.

   The reviewers recognize the ** theoretical rigor** in our proofs, and the **mechanistic insight** offered by our construction

   > **NFR7**: The appendices provide a **rigorous** "proof-by-construction." The mathematical links between specific attention mechanisms and algorithmic steps (like for power iteration) are **non-trivial and well-executed**.

   > **jojL**: two-stage architecture ... offers **rare mechanistic interpretability** and provides insight into how Transformers may implement semi-supervised reasoning.

   > **TXiC**: E2E-ICL is both **mathematically well-justified** and **empirically effective** at semi-supervised ICL.

   > **w9xM**: This paper proposed method that assigned different functional roles to different Transformer blocks, eg, representation extraction and ICL+categorial mapping. Such separation may **provide insights** into understanding the roles of different Transformer layers.


3. Empirically, across a wide range of data types, from synthetic manifolds to real images, we show that our model is highly label-efficient and **consistently outperforms strong baselines**, including a **standard Transformer with an order-of-magnitude more parameters**. Furthermore, we show that the unconstrained baseline Transformer learns representations which **align with our model**, highlighting the value of our construction as a **reference model** for understanding standard Transformers.

   The reviewers recognize the **rigor and thoroughness of our experiments**
   > **jojL:** Empirically, the method demonstrates strong label efficiency and **outperforms competitive baselines** across synthetic manifolds, product manifolds, and ImageNet100, showing **promising generalization and transfer**, especially under low-label regimes.

   > **TXiC**: **thoroughly explores** semi-supervised ICL, presenting experiments on **carefully designed** synthetic data, using multiple random seeds, and reporting **interesting ablations**.

   > **NFR7**: The method is shown to be highly effective, **outperforming strong baselines**, which validates the authors' algorithmic choices.

---

> ### Author Response · Authors · 2025-12-03
> **Rebuttal Summary (2/5 Response to NFR7)**
>
> ## Response to Questions and Concerns
>
> We now address the main concerns raised by the four reviewers. We organize the points by reviewer. In this summary, we omit minor questions related to clarifications of notation/mathematics, discussion of these can be found in our responses to each individual reviewer.
>
> ---
>
> ## Reviewer NFR7
>
> Reviewer **NFR7**'s main concern is about the necessity of **non-linear spectral method** and **iterative algorithm** in our construction:
>
> > **NFR7:** The paper ... doesn't first establish why this (sophisticated non-linear SSL algorithm) complexity is needed...A major question is whether a linear spectral method (e.g., PCA) would be sufficient.
> > **NFR7:** Your model relies on iterative algorithms (power iteration, GD) implemented via depth/loops. What are the theoretical trade-offs ... versus a hypothetical non-iterative (e.g., single-layer) but perhaps very wide model?
>
> ◆ We prove the necessity of **non-linear** spectral method using a **theoretical counterexample** of two concentric circles. We clarify that non‑linear spectral algorithms are **inherently iterative** (e.g., power iteration for eigenvectors, gradient descent–style refinement), and our architecture mirrors this structure inside the Transformer depth.
>
> ◆ We perform an additional empirical ablation, where we show  that performance degrades if (i) we use linear attention or (ii) we use a shallow-but-wide network.
>
> Detailed discussion [here](https://openreview.net/forum?id=lqrpmqrTnH&noteId=rzmTGHh6ow).
>
> ## To Area Chair:
>
> Reviewer NFR7 agreed that the above counterexample and ablation successfully addressed their concern, and **increased their score to 6**:
>
> >  This successfully addresses my primary concern regarding the justification for the architectural complexity and highlights the unique value of your constructive approach. In light of this strong theoretical defense and the empirical validation, I am raising my score to 6.
>
>
> Furthermore, reviewer NFR7 indicated willingness **increase their score to 8** if we address reviewer w9xM's concerns about **related work and model complexity**
> >  I also note the valid points raised by Reviewer w9xM, particularly regarding the discussion of related/concurrent works and clarifications on model complexity... If those concerns are resolved satisfactorily, I am open to further increasing my score to 8.

---

> ### Author Response · Authors · 2025-12-03
> **Rebuttal Summary (3/5 Response to w9xM)**
>
> ## Reviewer w9xM
>
> Reviewer w9xM has two main concerns: (i) our model size scaling with $O(n^2)$, and (ii) how our results compare with related work [1,2,3].
>
> ---
>
> ### Model size and dependence on $n$
>
> > **w9xM:** According to Appendix A.1, due to such input construction, the number of model parameters have increased from $d \times d$ to $n \times n$, which **greatly increases the computational and memory cost**... Due to the Weaknesses 2, the authors could only test on the fixed in-context sample size (n = 100).
>
> This concern arises from a misunderstanding of our implementation and our construction. We clarify that
>
> ◆ Our **parameter count** scales only as $O(n)$, same as **standard Transformers**, whose parameterization is  **also $O(n)$ once positional encodings are included**.
>
> ◆ Our **memory cost and computational complexity** scale as $O(n^2)$, again same as **standard Transformers**.
>
> ◆ Our choice of $n=100$ was for **consistent experimental comparison** across different methods, **not** a fundamental limit of our model.
>
> See [here](https://openreview.net/forum?id=lqrpmqrTnH&noteId=hRNDQGlBia) for a detailed explanation. Thus this concern should be **fully resolved**: our (i) parameter count, (ii) computational complexity and (iii) memory complexity are no more than **standard Transformers**.
>
> ---
>
> ### Positioning relative to recent semi-/unsupervised ICL work
>
> > **w9xM:** There are numerous prior works that have explored semi-supervision ICL. For example [1,2,3].
>
> > **TXiC:** It would be nice to at least mention [1] (recent empirical work on unsupervised ICL).
>
> We have now added a **dedicated discussion** of [1,2,3] in Appendix I (to be merged into the related work section in the camera-ready).
>
> * [1,2] are **empirical** studies showing that unlabelled data can substantially improve in-context learning performance in large language models. Our work is **complementary**: we provide a **constructive, mechanistic theory** of how a Transformer can implement in-context *semi-supervised* learning from first principles, under a manifold hypothesis.
> * [3] is theoretical, but focuses on a specific **binary Gaussian mixture model**. In contrast, our analysis targets a **broader family of manifold-structured problems**. Moreover, [3] is **contemporaneous**, per ICLR rules, with our submission.
>
> See [here](https://openreview.net/forum?id=lqrpmqrTnH&noteId=hRNDQGlBia) for a detailed discussion.
>
>
>
>
>
> [1] [Agarwal et al. 2024] Many-shot in-context learning
>
> [2] [Chen et al., 2025] Maple: Many-shot adaptive pseudo-labeling for in-context learning.
>
> [3] [Li et al., 2025) When and How Unlabeled Data Provably Improve In-Context Learning
>
>
> ---
>
> ## To Area Chair:
>
> Reviewer **w9xM**’s two main concerns are both **factual/clarificatory** rather than fundamental:
>
> 1. **Model size / dependence on $n$** — stemmed from a misunderstanding of our parameterization; we now [clearly explain](https://openreview.net/forum?id=lqrpmqrTnH&noteId=hRNDQGlBia) that our constructions are **independent of $n$** and our implementation is **linear in $n$**, with concrete parameter tables.
> 2. **Related and concurrent work** — has been addressed by a detailed discussion of [1,2,3], clearly positioning our work as a **complementary mechanistic theory** rather than an empirical study of LLMs.
>
> Given that:
>
> * these points have **concrete, unambiguous answers**, and
> * NFR7 **explicitly stated** they are willing to **raise their score to 8** once w9xM’s concerns about model complexity and related work are resolved,
>
> We believe it should be clear that these issues are adequately addressed in the revision and do not reflect substantive weaknesses of the paper.

---

> ### Author Response · Authors · 2025-12-03
> **Rebuttal Summary (4/5 Response to jojL)**
>
> ## Reviewer jojL
>
> Reviewer jojL has three main concerns: (i) our experiments are limited to **small, controlled episodic settings**, (ii) our method relies on **strong geometry‑driven inductive bias**, (iii) use of non‑standard attention variants and customized modules increases **architectural complexity and training sensitivity**.
>
> ### Scale and scope of experiments
>
> > **jojL:** experimental evaluation is limited in scale and focuses primarily on small episodic settings, leaving open questions about robustness in large, noisy, or highly diverse real-world scenarios.
>
> ◆ Our goal is **mechanistic understanding** of in‑context semi‑supervised learning (IC‑SSL). Controlled episodic setups enable us to *isolate and study the underlying mechanism* and are **standard in numerous existing mechanistic studies of ICL** (see detailed discussion [here](https://openreview.net/forum?id=lqrpmqrTnH&noteId=efj3fhbdFA))
>
> ◆ Our experiments span a **wide range** of difficulty and data types: (i) synthetic low‑dimensional manifolds  (ii) higher‑dimensional product manifolds (iii) diffusion‑generated image manifolds in pixel space (iv) ImageNet100 manifolds in VGG feature space.
>
> ◆ Across the above setups, we **consistently verify our proposed "representation learning + ICL" mechanism**.
>
> ---
>
> ### Geometry‑driven inductive bias / manifold assumption
>
> > **jojL:** The model relies on a strong geometry-driven inductive bias, which may restrict flexibility when the data does not exhibit clear manifold structure.
>
> ◆ The geometry‑aware inductive bias lets us **connect** the Transformer to classical manifold‑based SSL and obtain a **transparent mechanism**.
>
> ◆ Our construction **outperforms** a fully unconstrained baseline Transformer on semi‑supervised image tasks (diffusion manifolds and ImageNet100) despite using much fewer parameters (Figs. 3 and 5).
>
> ◆ Fig. 3 and Table 1 show that the baseline Transformer learns representations strongly aligned with our model's learned representation, indicating that our model is a **useful mechanistic proxy** for standard Transformers.
>
> ◆ **Learned feature spaces** often do satisfy the manifold assumption. Our ImageNet100 experiment (which uses VGG features) suggests that our model can be **composed with standard representation‑learning backbones**.
>
> ---
>
> ### “Non‑standard” attention, complexity, and training sensitivity
>
> > **jojL:** the architecture requires non-standard attention variants and customized modules that could hinder practical adoption. Additionally, the overall complexity introduces training sensitivity and may limit applicability beyond the studied domains.
>
> ◆ Implementation‑wise, we use a **standard Transformer trained end‑to‑end** (Sec. 3), with two differences:
>
> 1. **Structured sparsity** on Q/K/V (block‑diagonal/diagonal), which *reduces* the number of parameters.
> 2. **Alternative attention kernels**: $Attn_{linear}$ has been widely studied and used in large‑scale LMs, $Attn_{rbf}$ is **equivalent** to standard softmax attention under **layer normalization**.
>
> ◆ During training, a **single Transformer with standard layers** is trained **end-to-end**.
>
> ◆ Empirically, we **did not observe training sensitivity** in any of our experiments, across all seeded runs (note the small error bars in our experiments). The added sparsity makes training **more stable** than the baseline Transformer (Figure 3)
>
> ---
>
> ### To Area Chair
>
> jojL’s comments mainly concern **scope and inductive biases**, rather than flaws in our results:
>
> 1. The “limited scale” point reflects a deliberate choice of **controlled mechanistic experiments**, which is standard in numerous [existing mechanistic studies](https://openreview.net/forum?id=lqrpmqrTnH&noteId=MgcgL1taiy) of Transformers/ICL and is complementary to recent large‑scale empirical IC‑SSL studies.
> 2. The **geometry‑driven bias and attention parameterizations** are design choices that make the mechanism analyzable; empirically, they are **aligned with and even outperform** unconstrained Transformers and are implemented entirely with **standard Transformer components**.
> 3. Our architecture and training are **not more complex** than a standard Transformer. We did not observe any additional **training sensitivity**.
>
> We therefore view these as questions about **generality and future extensions**, not substantive weaknesses of our theoretical or experimental claims.

---

> ### Author Response · Authors · 2025-12-03
> **Rebuttal Summary (5/5 Response to TXiC)**
>
> ## Reviewer TXiC
> Reviewer TXiC had few concerns and viewed our work positively.
>
> > It would be nice to at least mention recent empirical work [1] on unsupervised ICL.
>
> ◆ We refer to the discussion under w9xM above, who also referenced [1].
>
> > The claim in line 343 that the model is competitive on real data seems a bit ill-supported, as even the ImageNet task is a fairly synthetic classification task.
>
> ◆ We revised the wording to be more precise: we now emphasize that our experiments use **real-world image data** (ImageNet), but that the task itself—semi-supervised image classification—is best viewed as an **artificial benchmark**.
>
> > Can you provide the parameter counts for all model types trained? It's a bit hard to tell from the descriptions alone whether some methods might have more learnable expressivity.
>
> ◆ We provide this in appendix $H$. Notably, our model ($\approx 10^4$ parameters) outperforms the **baseline Transformer** ($\approx 10^6$ parameters) despite using **substantially fewer** parameters, directly addressing the concern that performance gains might simply reflect larger capacity.

---

### Meta-Review · Area_Chair_CxD1 · 2026-01-01

**Summary:**

The paper introduces and studies in-context semi-supervised learning (IC-SSL),  which extends Transformer-based in-context learning to settings with few labeled and many unlabeled examples. The authors study a semi-supervised in-context learning setup, where inputs include multiple examples with only a few gold-labeled.  The reviewers find that the paper proposed a novel semi‑supervised extension of in‑context learning. The proposed two‑stage architecture provides mechanistic interpretability and insights into how Transformers may implement semi‑supervised reasoning. Empirically, the method demonstrates strong label efficiency and outperforms competitive baselines on synthetic manifolds, product manifolds, and ImageNet100, showing promising generalization and transfer, especially in low‑label regimes. The reviewers' concerns primarily center on insufficient experimental validation and imprecise claims.
 Overall, the work is both theoretically well‑motivated and empirically thorough, supported by carefully designed experiments and extensive ablations.

The authors provided a thorough rebuttal and successfully gained the reviewers' support (NFR7 recommended raising the score from 4 to 8, based on their assessment that the authors had adequately addressed the concerns raised by another reviewer). Therefore, taking into account the reviewers' final scores and the productive discussion between reviewers and authors, I support accepting this paper.

**Reviewer Concerns:**

The primary concerns raised by reviewer NFR7 has been addressed.
I believe the conerns of Reviewer w9xM, particularly regarding the discussion of related/concurrent works and clarifications on model complexity, have also been addressed. The conerns from the other reviewers has also been addressed to some extent.

**Reviewer Scores:**

NFR7 recommended raising the score from 4 to 8, based on their assessment that the authors had adequately addressed the concerns raised by another reviewer (w9xM).

I thikn reviewer w9xM may also raise his/her score.

---

### Decision · Program_Chairs · 2026-01-26

Accept (Poster)